# Humans strategically shift decision bias by flexibly adjusting sensory evidence accumulation

Niels A Kloosterman[1,2]*, Jan Willem de Gee[3,4], Markus Werkle-Bergner[2],
Ulman Lindenberger[1,2], Douglas D Garrett[1,2†], Johannes Jacobus Fahrenfort[4,5†]

[1]Max Planck UCL Centre for Computational Psychiatry and Ageing Research, Max Planck Institute for Human Development, Berlin, Germany; [2]Center for Lifespan Psychology, Max Planck Institute for Human Development, Berlin, Germany; [3]Department of Neurophysiology and Pathophysiology, University Medical Center Hamburg-Eppendorf, Hamburg, Germany; [4]Department of Psychology, University of Amsterdam, Amsterdam, The Netherlands; [5]Department of Experimental and Applied Psychology, Vrije Universiteit, Amsterdam, The Netherlands

*For correspondence:
kloosterman@mpib-berlin.mpg.de

†These authors contributed equally to this work

Competing interests: The authors declare that no competing interests exist.

**Abstract** Decision bias is traditionally conceptualized as an internal reference against which sensory evidence is compared. Instead, we show that individuals implement decision bias by shifting the rate of sensory evidence accumulation toward a decision bound. Participants performed a target detection task while we recorded EEG. We experimentally manipulated participants' decision criterion for reporting targets using different stimulus-response reward contingencies, inducing either a liberal or a conservative bias. Drift diffusion modeling revealed that a liberal strategy biased sensory evidence accumulation toward target-present choices. Moreover, a liberal bias resulted in stronger midfrontal pre-stimulus 2—6 Hz (theta) power and suppression of pre-stimulus 8—12 Hz (alpha) power in posterior cortex. Alpha suppression in turn was linked to the output activity in visual cortex, as expressed through 59—100 Hz (gamma) power. These findings show that observers can intentionally control cortical excitability to strategically bias evidence accumulation toward the decision bound that maximizes reward.
DOI: https://doi.org/10.7554/eLife.37321.001

## Introduction

Perceptual decisions arise not only from the evaluation of sensory evidence, but are often biased toward a given choice alternative by environmental factors, perhaps as a result of task instructions and/or stimulus-response reward contingencies (*White and Poldrack, 2014*). The ability to willfully control decision bias could potentially enable the behavioral flexibility required to survive in an ever-changing and uncertain environment. But despite its important role in decision making, the neural mechanisms underlying decision bias are not fully understood.

The traditional account of decision bias comes from signal detection theory (SDT) (*Green and Swets, 1966*). In SDT, decision bias is quantified by estimating the relative position of a decision point (or 'criterion') in between sensory evidence distributions for noise and signal (see *Figure 1A*). In this framework, a more liberal decision bias arises by moving the criterion closer toward the noise distribution (see green arrow in *Figure 1A*). Although SDT has been very successful at quantifying decision bias, how exactly bias affects decision making and how it is reflected in neural activity remains unknown. One reason for this lack of insight may be that SDT does not have a temporal component to track how decisions are reached over time (*Fetsch et al., 2014*). As an alternative to SDT, the drift diffusion model (DDM) conceptualizes perceptual decision making as the accumulation

**eLife digest** How do you decide whether to buy a new car? One factor to consider is how well the economy is doing. During an economic boom, you might happily commit to buying a new vehicle that goes on sale, but prefer to sit on your savings during a financial crisis, despite how good the offer may be. Adjusting how you make decisions in situations like this can help you optimize choices in an ever-changing world.

It's currently thought that when deciding, we accumulate evidence for each of the available options. When evidence for one of the options passes a threshold, we choose that option. External factors – such as a booming economy when considering buying a car – could bias this process in two different ways. The standard view is that they move the starting point of evidence accumulation towards one of the two choices, so that the threshold for choosing that option is more easily reached. Alternatively, they could bias the accumulation process itself, so that evidence builds up more quickly towards one of the choices.

To distinguish between these possibilities, Kloosterman et al. asked volunteers to press a button whenever they detected a target hidden among a stream of visual patterns. To bias their decisions, volunteers were penalized differently in two experimental conditions: either when they failed to report a target (a 'miss'), or when they 'detected' a target when in fact nothing was there (a 'false alarm'). As expected, punishing participants for missing a target made them more liberal towards reporting targets, whereas penalizing false alarms made them more conservative.

Computational modeling of behavior revealed that when participants used a liberal strategy, they did not move closer to the threshold for deciding target presence. Instead, they accumulated evidence for target presence at a faster rate, even when in fact no target was shown. Brain activity recorded during this task reveals how this bias in evidence accumulation might come about. When a volunteer adopted a liberal response strategy, visual brain areas showed a reduction in low-frequency 'alpha' waves, suggesting increased attention. This in turn triggered an increase in high-frequency 'gamma' waves, reflecting biased evidence accumulation for target presence (irrespective of whether a target actually appeared or not).

Overall, the findings reported by Kloosterman et al. suggest that we can strategically bias perceptual decision-making by varying how quickly we accumulate evidence in favor of different response options. This might explain how we are able to adapt our decisions to environments that differ in payoffs and punishments. The next challenge is to understand whether such biases also affect high-level decisions, for example, when purchasing a new car.

DOI: https://doi.org/10.7554/eLife.37321.002

of noisy sensory evidence over time into an internal decision variable (*Bogacz et al., 2006*; *Gold and Shadlen, 2007*; *Ratcliff and McKoon, 2008*). A decision in this model is made when the decision variable crosses one of two decision bounds corresponding to the choice alternatives. After one of the bounds is reached, the corresponding decision can subsequently either be actively reported, e.g. by means of a button press indicating a detected signal, or it could remain without behavioral report when no signal is detected (*Ratcliff et al., 2018*). Within this framework, a strategic decision bias imposed by the environment can be modelled in two different ways: either by moving the starting point of evidence accumulation closer to one of the boundaries (see green arrow in *Figure 1B*), or by biasing the rate of the evidence accumulation process itself toward one of the boundaries (see green arrow in *Figure 1C*). In both the SDT and DDM frameworks, decision bias shifts have little effect on the sensitivity of the observer when distinguishing signal from noise; they predominantly affect the relative response ratios (and in the case of DDM, the speed with which one or the other decision bound is reached). There has been some evidence to suggest that decision bias induced by shifting the criterion is best characterized by a drift bias in the DDM (*Urai et al., 2018*; *White and Poldrack, 2014*). However, the drift bias parameter has as yet not been related to a well-described neural mechanism.

Regarding the neural underpinnings of decision bias, there have been a number of reports about a correlational relationship between cortical population activity measured with EEG and decision bias. For example, spontaneous trial-to-trial variations in pre-stimulus oscillatory activity in the 8—12

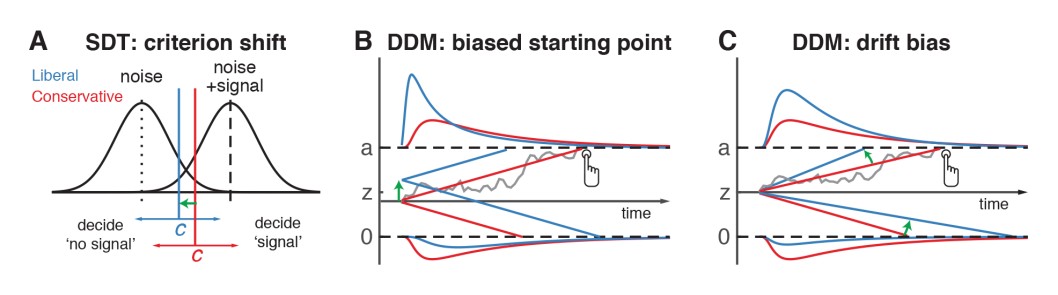

**Figure 1.** Theoretical accounts of decision bias. (**A**) Signal-detection-theoretic account of decision bias. Signal and noise + signal distributions are plotted as a function of the strength of internal sensory evidence. The decision point (or criterion) that determines whether to decide signal presence or absence is plotted as a vertical criterion line *c*, reflecting the degree of decision bias. *c* can be shifted left- or rightwards to denote a more liberal or conservative bias, respectively (green arrow indicates a shift toward more liberal). (**B, C**) Drift diffusion model (DDM) account of decision bias, in which decisions are modelled in terms of a set of parameters that describe a dynamic process of sensory evidence accumulation toward one of two decision bounds. When sensory input is presented, evidence starts to accumulate (drift) over time after initialization at the starting point *z*. A decision is made when the accumulated evidence either crosses decision boundary *a* (signal presence) or decision boundary *0* (no signal). After a boundary is reached, the corresponding decision can be either actively reported by a button press (e.g. for signal-present decisions), or remain implicit, without a response (for signal-absent decisions). The DDM can capture decision bias through a shift of the starting point of the evidence accumulation process (panel B) or through a shift in bias in the rate of evidence accumulation toward the different choices (panel C). These mechanisms are dissociable through their differential effect on the shape of the reaction time (RT) distributions, as indicated by the curves above and below the graphs for target-present and target-absent decisions, respectively. Panels B. and C. are modified and reproduced with permission from *Urai et al., 2018* (Figure 1, published under a CC BY 4.0 license).

DOI: https://doi.org/10.7554/eLife.37321.003

Hz (alpha) band have been shown to correlate with decision bias and confidence (*Iemi and Busch, 2018*; *Limbach and Corballis, 2016*). Alpha oscillations, in turn, have been proposed to be involved in the gating of task-relevant sensory information (*Jensen and Mazaheri, 2010*), possibly encoded in high-frequency (gamma) oscillations in visual cortex (*Ni et al., 2016*; *Popov et al., 2017*). Although these reports suggest links between pre-stimulus alpha suppression, sensory information gating, and decision bias, they do not uncover whether pre-stimulus alpha plays an instrumental role in decision bias and how exactly this might be achieved. Specifically, it is unknown whether an experimentally induced shift in decision bias is implemented in the brain by willfully adjusting pre-stimulus alpha in sensory areas.

Here, we explicitly investigate these potential mechanisms by employing a task paradigm in which shifts in decision bias were experimentally induced within participants through (a) instruction and (b) asymmetries in stimulus-response reward contingencies during a visual target detection task. By applying drift diffusion modeling to the participants' choice behavior, we show that the effect of strategically adjusting decision bias is best captured by the drift bias parameter, which is thought to reflect a bias in the rate of sensory evidence accumulation toward one of the two decision bounds. To substantiate a neural mechanism for this effect, we demonstrate that this bias shift is accompanied by changes in pre-stimulus midfrontal 2–6 Hz (theta) power, as well as changes in sensory alpha suppression. Pre-stimulus alpha suppression in turn is linked to the post-stimulus output of visual cortex, as reflected in gamma power modulation. Critically, we show that gamma activity accurately predicted the strength of evidence accumulation bias within participants, providing a direct link between the proposed mechanism and decision bias. Together, these findings identify a neural mechanism by which intentional control of cortical excitability is applied to strategically bias perceptual decisions in order to maximize reward in a given ecological context.

## Results

### Manipulation of decision bias affects sensory evidence accumulation

In three EEG recording sessions, human participants (N = 16) viewed a continuous stream of horizontal, vertical and diagonal line textures alternating at a rate of 25 textures/second. The participants' task was to detect an orientation-defined square presented in the center of the screen and report it via a button press (*Figure 2A*). Trials consisted of a fixed-order sequence of textures embedded in the continuous stream (total sequence duration 1 s). A square appeared in the fifth texture of a trial in 75% of the presentations (target trials), while in 25% a homogenous diagonal texture appeared in the fifth position (nontarget trials). Although the onset of a trial within the continuous stream of textures was not explicitly cued, the similar distribution of reaction times in target and nontarget trials suggests that participants used the temporal structure of the task even when no target appeared (*Figure 2—figure supplement 1A*). Consistent and significant EEG power modulations after trial onset (even for nontarget trials) further confirm that subjects registered trial onsets in the absence of an explicit cue, plausibly using the onset of a fixed order texture sequence as an implicit cue (*Figure 2—figure supplement 1B*).

In alternating nine-minute blocks of trials, we actively biased participants' perceptual decisions by instructing them either to report as many targets as possible ('Detect as many targets as possible!"; liberal condition), or to only report high-certainty targets ("Press only if you are really certain!"; conservative condition). Participants were free to respond at any time during a block whenever they detected a target. A trial was considered a target present response when a button press occurred before the fixed-order sequence ended (i.e. within 0.84 s after onset of the fifth texture containing the (non)target, see *Figure 2A*). We provided auditory feedback and applied monetary penalties following missed targets in the liberal condition and following false alarms in the conservative condition (*Figure 2A*; see Materials and methods for details). The median number of trials for each SDT category across participants was 1206 hits, 65 false alarms, 186 misses and 355 correct rejections in the liberal condition, and 980 hits, 12 false alarms, 419 misses and 492 correct rejections in the conservative condition.

Participants reliably adopted the intended decision bias shift across the two conditions, as shown by both the hit rate and the false alarm rate going down in tandem as a consequence of a more conservative bias (*Figure 2B*). The difference between hit rate and false alarm rate was not significantly modulated by the experimental bias manipulations (p=0.81, two-sided permutation test, 10,000 permutations, see *Figure 2B*). However, target detection performance computed using standard SDT $d'$ (perceptual sensitivity, reflecting the distance between the noise and signal distributions in *Figure 1A*) (*Green and Swets, 1966*) was slightly higher during conservative (liberal: $d'$=2.0 (s.d. 0.90) versus conservative: $d'$=2.31 (s.d. 0.82), p=0.0002, see *Figure 2C*, left bars). We quantified decision bias using the standard SDT criterion measure $c$, in which positive and negative values reflect conservative and liberal biases, respectively (see the blue and red vertical lines in *Figure 1A*). This uncovered a strong experimentally induced bias shift from the conservative to the liberal condition (liberal: $c = -0.13$ (s.d. 0.4), versus conservative: $c = 0.73$ (s.d. 0.36), p=0.0001, see *Figure 2C*), as well as a conservative average bias across the two conditions ($c = 0.3$ (s.d. 0.31), p=0.0013).

Because the SDT framework is static over time, we further investigated how bias affected various components of the dynamic decision process by fitting different variants of the drift diffusion model (DDM) to the behavioral data (*Figure 1B,C*) (*Ratcliff and McKoon, 2008*). The DDM postulates that perceptual decisions are reached by accumulating noisy sensory evidence toward one of two decision boundaries representing the choice alternatives. Crossing one of these boundaries can either trigger an explicit behavioral report to indicate the decision (for target-present responses in our experiment), or remain implicit (i.e. without active response, for target-absent decisions in our experiment). The DDM captures the dynamic decision process by estimating parameters reflecting the rate of evidence accumulation (drift rate), the separation between the boundaries, as well as the time needed for stimulus encoding and response execution (non-decision time) (*Ratcliff and McKoon, 2008*). The DDM is able to estimate these parameters based on the shape of the RT distributions for actively reported (target-present) decisions along with the total number of trials in which no response occurred (i.e. implicit target-absent decisions) (*Ratcliff et al., 2018*).

We fitted two variants of the DDM to distinguish between two possible mechanisms that can bring about a change in choice bias: one in which the starting point of evidence accumulation moves

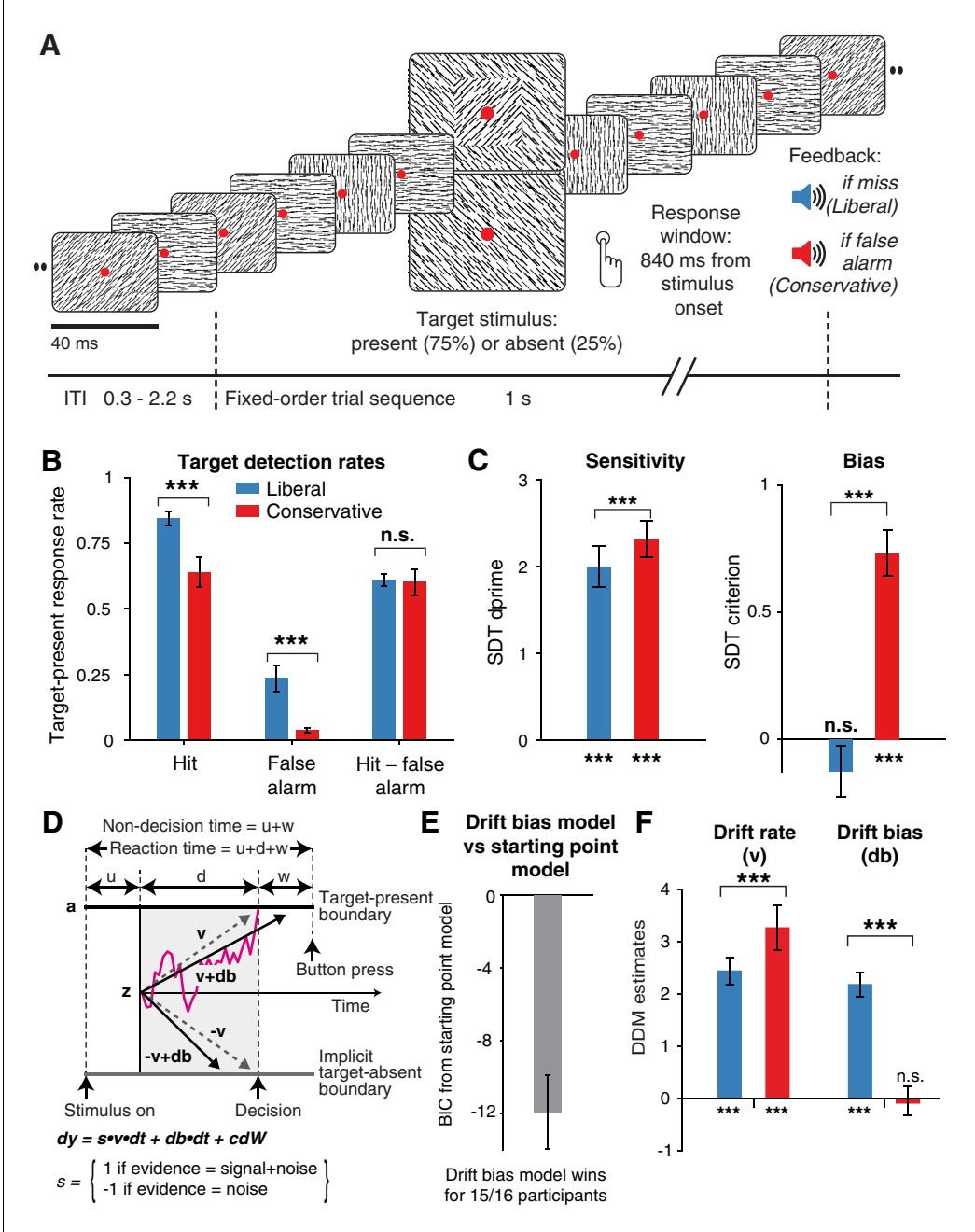

**Figure 2.** Strategic decision bias shift toward liberal biases evidence accumulation. (A) Schematic of the visual stimulus and task design. Participants viewed a continuous stream of full-screen diagonally, horizontally and vertically oriented textures at a presentation rate of 40 ms (25 Hz). After random inter-trial intervals, a fixed-order sequence was presented embedded in the stream. The fifth texture in each sequence either consisted of a single diagonal orientation (target absent), or contained an orthogonal orientation-defined square (either 45° or 135° orientation). Participants decided whether they had just seen a target, reporting detected targets by button press. Liberal and conservative conditions were administered in alternating nine-min blocks by penalizing either misses or false alarms, respectively, using aversive tones and monetary deductions. Depicted square and fixation dot sizes are not to scale. (B) Average detection rates (hits and false alarms) during both conditions. Miss rate is equal to 1 – hit rate since both are computed on stimulus present trials, and correct-rejection rate is equal to 1 – false alarm rate since both are computed on stimulus absent trials, together yielding the four SDT stimulus-response categories. (C) SDT parameters for sensitivity and criterion. (D) Schematic and simplified equation of drift diffusion model accounting for reaction time distributions for actively reported target-present and implicit target-absent decisions. Decision bias in this model can be implemented by either shifting the starting point of the evidence

*Figure 2 continued on next page*

*Figure 2 continued*

accumulation process (Z), or by adding an evidence-independent constant ('drift bias', db) to the drift rate. See text and *Figure 1* for details. Notation: dy, change in decision variable y per unit time dt; v·dt, mean drift (multiplied with one for signal + noise (target) trials, and −1 for noise-only (nontarget) trials); db·dt, drift bias; and cdW, Gaussian white noise (mean = 0, variance = c2·dt). (E) Difference in Bayesian Information Criterion (BIC) goodness of fit estimates for the drift bias and the starting point models. A lower delta BIC value indicates a better fit, showing superiority of the drift bias model to account for the observed results. (F) Estimated model parameters for drift rate and drift bias in the drift bias model. Error bars, SEM across 16 participants. ***p<0.001; n.s., not significant. Panel D. is modified and reproduced with permission from *de Gee et al. (2017)* (Figure 4A, published under a CC BY 4.0 license).

DOI: https://doi.org/10.7554/eLife.37321.004

The following source data and figure supplements are available for figure 2:

**Source data 1.** This csv table contains the data for *Figure 2* panels B, C, E and F.

DOI: https://doi.org/10.7554/eLife.37321.008

**Figure supplement 1.** Behavioral and neurophysiological evidence that participants were sensitive to the implicit task structure.

DOI: https://doi.org/10.7554/eLife.37321.005

**Figure supplement 2.** Single-participant drift diffusion model fits for the drift bias and starting point models for both conditions.

DOI: https://doi.org/10.7554/eLife.37321.006

**Figure supplement 3.** Signal-detection-theoretic (SDT) behavioral measures during both conditions correspond closely to drift diffusion modeling (DDM) parameters.

DOI: https://doi.org/10.7554/eLife.37321.007

closer to one of the decision boundaries ('starting point model', *Figure 1B*) (*Mulder et al., 2012*), and one in which the drift rate itself is biased toward one of the boundaries (*de Gee et al., 2017*) ('drift bias model', see *Figure 1C*, referred to as drift criterion by *Ratcliff and McKoon (2008)*). The drift bias parameter is determined by estimating the contribution of an evidence-independent constant added to the drift (*Figure 2D*). In the two respective models, we freed either the drift bias parameter (db, see *Figure 2D*) for the two conditions while keeping starting point (z) fixed across conditions (for the drift bias model), or vice versa (for the starting point model). Permitting only one parameter at a time to vary freely between conditions allowed us to directly compare the models without having to penalize either model for the number of free parameters. These alternative models make different predictions about the shape of the RT distributions in combination with the response ratios: a shift in starting point results in more target-present choices particularly for short RTs, whereas a shift in drift bias grows over time, resulting in more target-present choices also for longer RTs (*de Gee et al., 2017*; *Ratcliff and McKoon, 2008*; *Urai et al., 2018*). The RT distributions above and below the evidence accumulation graphs in *Figure 1B and C* illustrate these different effects. In both models, all of the non-bias related parameters (drift rate v, boundary separation a and non-decision time u + w, see *Figure 2D*) were also allowed to vary by condition.

We found that the starting point model provided a worse fit to the data than the drift bias model (starting point model, Bayesian Information Criterion (BIC) = 7938; drift bias model, BIC = 7926, *Figure 2E*, see Materials and methods for details). Specifically, for 15/16 participants, the drift bias model provided a better fit than the starting point model, for 12 of which delta BIC >6, indicating strong evidence in favor of the drift bias model (*Kass and Raftery, 1995*). Despite the lower BIC for the drift bias model, however, we note that to the naked eye both models provide similarly reasonable fits to the single participant RT distributions (*Figure 2—figure supplement 2*). Finally, we compared these two models to a model in which both drift bias and starting point were fixed across the conditions, while still allowing the non-bias-related parameters to vary per condition. This model provided the lowest goodness of fit (delta BIC >6 for both models for all participants).

Given the superior performance of the drift bias model (in terms of BIC), we further characterized decision making under the bias manipulation using parameter estimates from this model (see below where we revisit the implausibility of the starting point model when inspecting the lack of pre-stimulus baseline effects in sensory or motor cortex). Drift rate, reflecting the participants' ability to discriminate targets and nontargets, was somewhat higher in the conservative compared to the liberal condition (liberal: v = 2.39 (s.d. 1.07), versus conservative: v = 3.06 (s.d. 1.16), p=0.0001,

permutation test, *Figure 2F*, left bars). Almost perfect correlations across participants in both conditions between DDM drift rate and SDT *d′* provided strong evidence that the drift rate parameter captures perceptual sensitivity (liberal, r = 0.98, p=1e$^{-10}$; conservative, r = 0.96, p=5e$^{-9}$, see *Figure 2—figure supplement 3A*). Regarding the DDM bias parameters, the condition-fixed starting point parameter in the drift bias model was smaller than half the boundary separation (i.e. closer to the target-absent boundary (z = 0.24 (s.d. 0.06), p<0.0001, tested against 0.5)), indicating an overall conservative starting point across conditions (*Figure 2—figure supplement 3D*), in line with the overall positive SDT criterion (see *Figure 2C*, right panel). Strikingly, however, whereas the drift bias parameter was on average not different from zero in the conservative condition (db = –0.04 (s.d. 1.17), p=0.90), drift bias was strongly positive in the liberal condition (db = 2.08 (s.d. 1.0), p=0.0001; liberal vs conservative: p=0.0005; *Figure 2F*, right bars). The overall conservative starting point combined with a condition-specific neutral drift bias explained the conservative decision bias (as quantified by SDT criterion) in the conservative condition (*Figure 2C*). Likewise, in the liberal condition, the overall conservative starting point combined with a condition-specific positive drift bias (pushing the drift toward the target-present boundary) explained the neutral bias observed with SDT criterion (*c* around zero for liberal, see *Figure 2C*).

Convergent with these modeling results, drift bias was strongly anti-correlated across participants with both SDT criterion (r = –0.89 for both conditions, p=4e$^{-6}$) and average reaction time (liberal, r = –0.57, p=0.02; conservative, r = –0.82, p=1e$^{-4}$, see *Figure 2—figure supplement 3B C*). The strong correlations between drift rate and *d′* on the one hand, and drift bias and *c* on the other, provide converging evidence that the SDT and DDM frameworks capture similar underlying mechanisms, while the DDM additionally captures the dynamic nature of perceptual decision making by linking the decision bias manipulation to the evidence accumulation process itself. As a control, we also correlated starting point with criterion, and found that the correlations were somewhat weaker in both conditions (liberal, r = –0.75.; conservative, r = –0.77), suggesting that the drift bias parameter better captured decision bias as instantiated by SDT.

Finally, the bias manipulation also affected two other parameters in the drift bias model that were not directly related to sensory evidence accumulation: boundary separation was slightly but reliably higher during the liberal compared to the conservative condition (p<0.0001), and non-decision time (comprising time needed for sensory encoding and motor response execution) was shorter during liberal (p<0.0001) (*Figure 2—figure supplement 3D*). In conclusion, the drift bias variant of the drift diffusion model best explained how participants adjusted to the decision bias manipulations. In the next sections, we used spectral analysis of the concurrent EEG recordings to identify a plausible neural mechanism that reflects biased sensory evidence accumulation.

## Task-relevant textures induce stimulus-related responses in visual cortex

Sensory evidence accumulation in a visual target detection task presumably relies on stimulus-related signals processed in visual cortex. Such stimulus-related signals are typically reflected in cortical population activity exhibiting a rhythmic temporal structure (*Buzsáki and Draguhn, 2004*). Specifically, bottom-up processing of visual information has previously been linked to increased high-frequency (>40 Hz, i.e. gamma) electrophysiological activity over visual cortex (*Bastos et al., 2015*; *Michalareas et al., 2016*; *Popov et al., 2017*; *van Kerkoerle et al., 2014*). *Figure 3* shows significant electrode-by-time-by-frequency clusters of stimulus-locked EEG power, normalized with respect to the condition-specific pre-trial baseline period (–0.4 to 0 s). We observed a total of four distinct stimulus-related modulations, which emerged after target onset and waned around the time of response: two in the high-frequency range (>36 Hz, *Figure 3A* (top) and *Figure 3B*) and two in the low-frequency range (<36 Hz, *Figure 3A* (bottom) and *Figure 3C*). First, we found a spatially focal modulation in a narrow frequency range around 25 Hz reflecting the steady state visual evoked potential (SSVEP) arising from entrainment by the visual stimulation frequency of our experimental paradigm (*Figure 3A*, bottom panel), as well as a second modulation from 42 to 58 Hz comprising the SSVEP's harmonic (*Figure 3A*, top panel). Both SSVEP frequency modulations have a similar topographic distribution (see left panels of *Figure 3A*).

Third, we observed a 59—100 Hz (gamma) power modulation (*Figure 3B*), after carefully controlling for high-frequency EEG artifacts due to small fixational eye movements (microsaccades) by removing microsaccade-related activity from the data (*Hassler et al., 2011*; *Hipp and Siegel, 2013*;

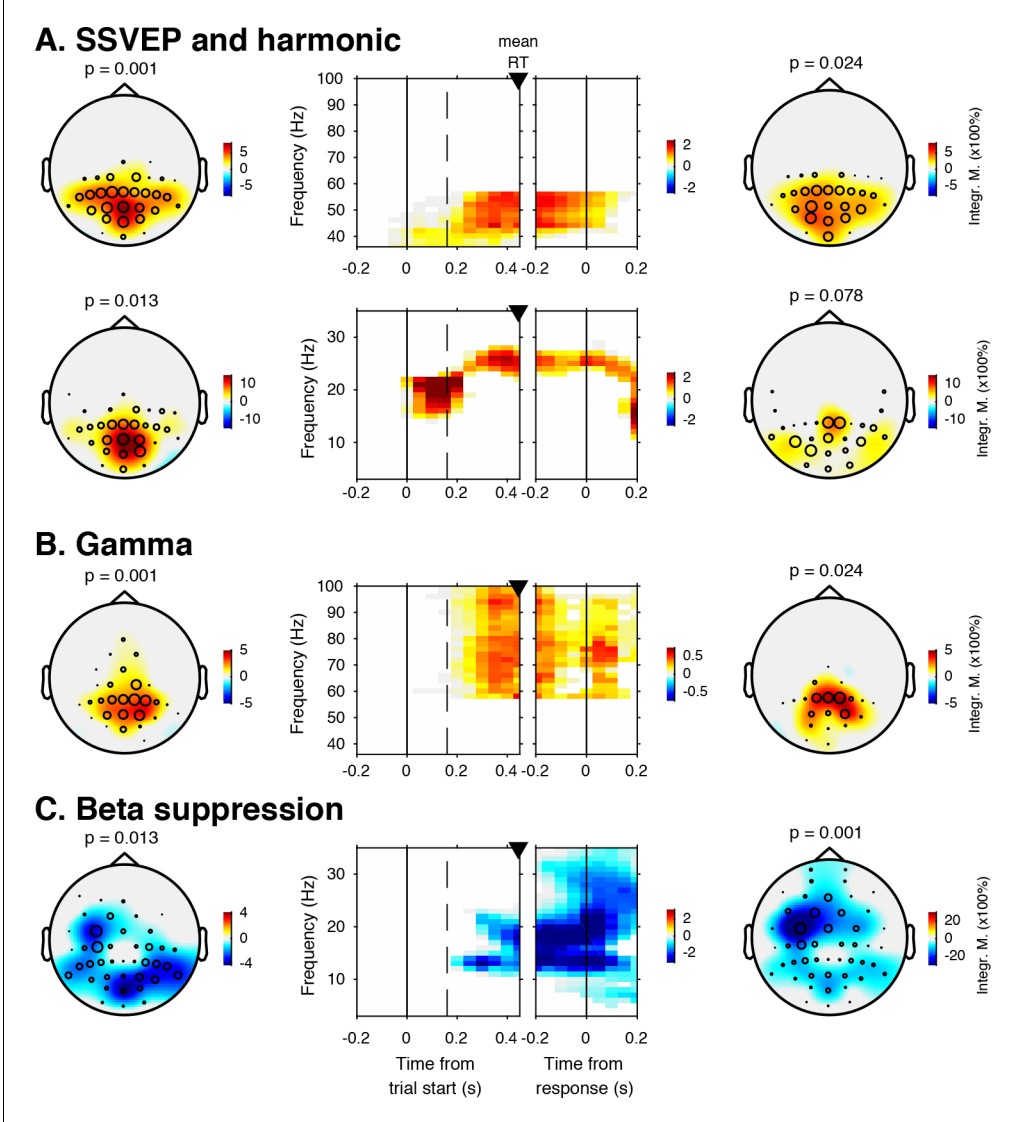

**Figure 3.** EEG spectral power modulations related to stimulus processing and motor response. Each panel row depicts a three-dimensional (electrodes-by-time-by-frequency) cluster of power modulation, time-locked both to trial onset (left two panels) and button press (right two panels). Power modulation outside of the significant clusters is masked out. Modulation was computed as the percent signal change from the condition-specific pre-stimulus period (–0.4 to 0 s) and averaged across conditions. Topographical scalp maps show the spatial extent of clusters by integrating modulation over time-frequency bins. Time-frequency representations (TFRs) show modulation integrated over electrodes indicated by black circles in the scalp maps. Circle sizes indicate electrode weight in terms of proportion of time-frequency bins contributing to the TFR. P-values above scalp maps indicate multiple comparison-corrected cluster significance using a permutation test across participants (two-sided, N = 14). Solid vertical lines indicate the time of trial onset (left) or button press (right), dotted vertical lines indicate time of (non)target onset. Integr. M., integrated power modulation. SSVEP, steady state visual evoked potential. (A) (Top) 42–58 Hz (SSVEP harmonic) cluster. (A) (Bottom). Posterior 23–27 Hz (SSVEP) cluster. (B) Posterior 59–100 Hz (gamma) cluster. The clusters in A (Top) and B were part of one large cluster (hence the same p-value), and were split based on the sharp modulation increase precisely in the 42–58 Hz range. (C) 12–35 Hz (beta) suppression cluster located more posteriorly aligned to trial onset, and more left-centrally when aligned to button press.

DOI: https://doi.org/10.7554/eLife.37321.009

*Yuval-Greenberg et al., 2008*), and by suppressing non-neural EEG activity through scalp current density (SCD) transformation (*Melloni et al., 2009*; *Perrin et al., 1989*) (see Materials and methods for details). Importantly, the topography of the observed gamma modulation was confined to posterior electrodes, in line with a role of gamma in bottom-up processing in visual cortex (*Ni et al., 2016*). Finally, we observed suppression of low-frequency beta (11—22 Hz) activity in posterior cortex, which typically occurs in parallel with enhanced stimulus-induced gamma activity (*Donner and Siegel, 2011*; *Kloosterman et al., 2015a*; *Meindertsma et al., 2017*; *Werkle-Bergner et al., 2014*) (*Figure 3C*). Response-locked, this cluster was most pronounced over left motor cortex (electrode C4), plausibly due to the right-hand button press that participants used to indicate target detection (*Donner et al., 2009*). In the next sections, we characterize these signals separately for the two conditions, investigating stimulus-related signals within a pooling of 11 occipito-parietal electrodes based on the gamma enhancement in *Figure 3B* (Oz, POz, Pz, PO3, PO4, and P1 to P6), and motor-related signals in left-hemispheric beta (LHB) suppression in electrode C4 (*Figure 3C*) (*O'Connell et al., 2012*).

## EEG power modulation time courses consistent with the drift bias model

Our behavioral results suggest that participants biased sensory evidence accumulation in the liberal condition, rather than changing their starting point. We next sought to provide converging evidence for this conclusion by examining pre-stimulus activity, post-stimulus activity, and motor-related EEG activity. Following previous studies, we hypothesized that a starting point bias would be reflected in a difference in pre-motor baseline activity between conditions before onset of the decision process (*Afacan-Seref et al., 2018*; *de Lange et al., 2013*), and/or in a difference in pre-stimulus activity such as in bottom up stimulus-related SSVEP and gamma power signals (*Figure 4A* shows the relevant clusters as derived from *Figure 3*). Thus, we first investigated the timeline of raw power in the SSVEP, gamma and LHB range between conditions (see *Figure 4B*). None of these markers showed a meaningful difference in pre-stimulus baseline activity. Statistically comparing the raw pre-stimulus activity between liberal and conservative in a baseline interval between –0.4 and 0 s prior to trial onset yielded p=0.52, p=0.51 and p=0.91, permutation tests, for the respective signals. This confirms a highly similar starting point of evidence accumulation in all these signals. Next, we predicted that a shift in drift bias would be reflected in a steeper slope of post-stimulus ramping activity (leading up to the decision). We reasoned that the best way of ascertaining such an effect would be to baseline the activity to the interval prior to stimulus onset (using the interval between –0.4 to 0 s), such that any post-stimulus effect we find cannot be explained by pre-stimulus differences (if any). The time course of post-stimulus and response-locked activity after baselining can be found in *Figure 4C*. All three signals showed diverging signals between the liberal and conservative condition after trial onset, consistent with adjustments in the process of evidence accumulation. Specifically, we observed higher peak modulation levels for the liberal condition in all three stimulus-locked signals (p=0.08, p=0.002 and p=0.023, permutation tests for SSVEP, gamma and LHB, respectively), and found a steeper slope toward the button press for LHB (p=0.04). Finally, the event related potential in motor cortex also showed a steeper slope toward report for liberal (p=0.07, *Figure 4*, bottom row, baseline plot is not meaningful for time-domain signals due to mean removal during preprocessing). Taken together, these findings provide converging evidence that participants implemented a liberal decision bias by adjusting the rate of evidence accumulation toward the target-present choice boundary, but not its starting point. In the next sections, we sought to identify a neural mechanism that could underlie these biases in the rate of evidence accumulation.

## Liberal bias is reflected in pre-stimulus midfrontal theta enhancement and posterior alpha suppression

Given a lack of pre-stimulus (starting-point) differences in specific frequency ranges involved in stimulus processing or motor responses (*Figure 4B*), we next focused on other pre-stimulus differences that might be the root cause of the post-stimulus differences we observed in *Figure 4C*. To identify such signals at high frequency resolution, we computed spectral power in a wide time window from –1 s until trial start. We then ran a cluster-based permutation test across all electrodes and frequencies in the low-frequency domain (1–35 Hz), looking for power modulations due to our experimental

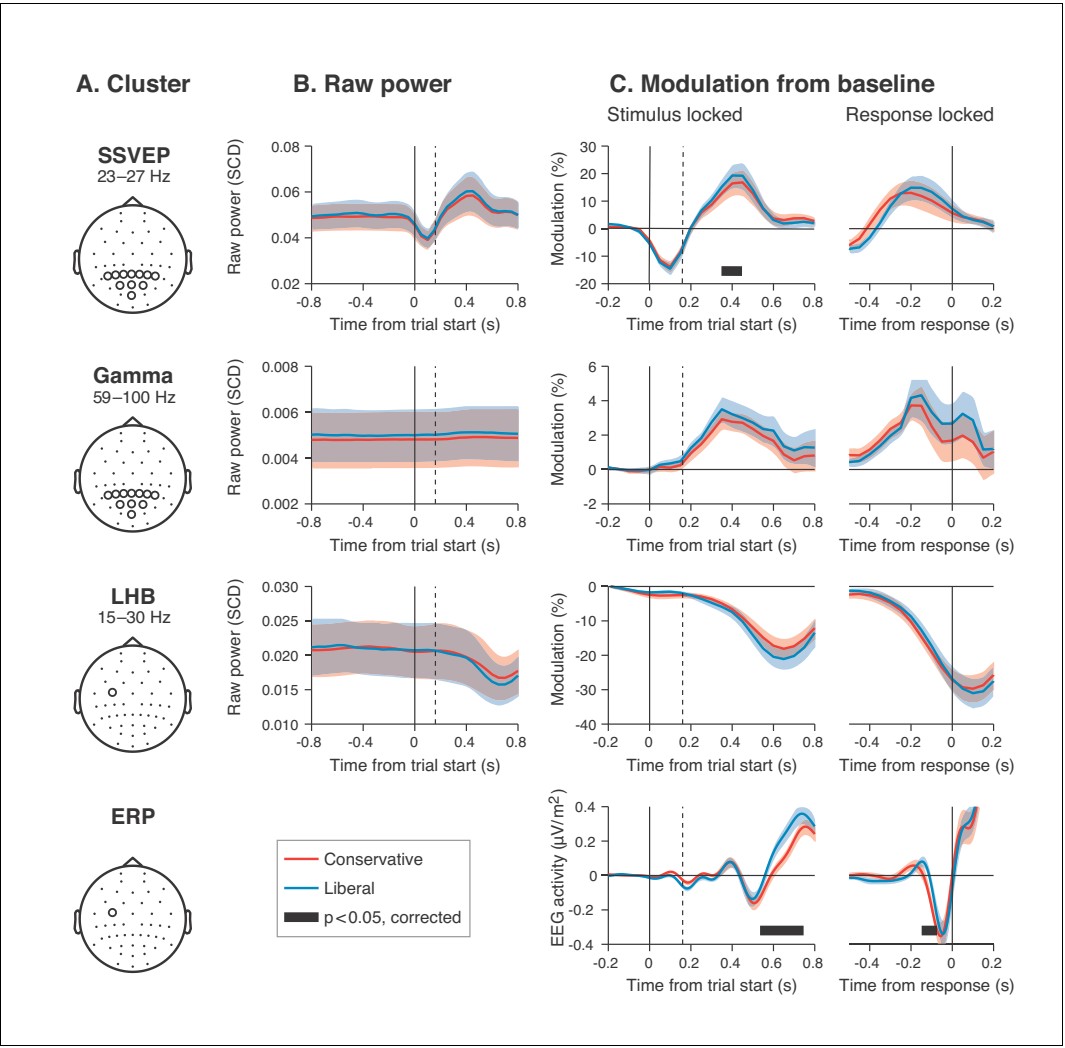

**Figure 4.** Experimental task manipulations affect the time course of stimulus- and motor-related EEG signals, but not its starting point. Raw power throughout the baseline period and time courses of power modulation time-locked to trial start and button press. (A) Relevant electrode clusters and frequency ranges (from *Figure 3*): Posterior SSVEP, Posterior gamma and Left-hemispheric beta (LHB). (B) The time course of raw power in a wide interval around the stimulus –0.8 to 0.8 s ms for these clusters. (C) Stimulus locked and response locked percent signal change from baseline (baseline period: –0.4 to 0 s). Error bars, SEM. Black horizontal bar indicates significant difference between conditions, cluster-corrected for multiple comparison (p<0.05, two sided). SSVEP, steady state visual evoked potential; LHB, left hemispheric beta; ERP, event-related potential; SCD, scalp current density.

DOI: https://doi.org/10.7554/eLife.37321.010

manipulations. Pre-stimulus spectral power indeed uncovered two distinct modulations in the liberal compared to the conservative condition: (1) theta modulation in midfrontal electrodes and (2) alpha modulation in posterior electrodes. *Figure 5A* depicts the difference between the liberal and conservative condition, confirming significant clusters (p<0.05, cluster-corrected for multiple comparisons) of enhanced theta (2–6 Hz) in frontal electrodes (Fz, Cz, FC1,and FC2), as well as suppressed alpha (8—12 Hz) in a group of posterior electrodes, including all 11 electrodes selected previously based on post-stimulus gamma modulation (*Figure 3*). The two modulations were uncorrelated across participants (r = 0.06, p=0.82), suggesting they reflect different neural processes related to our experimental task manipulations. These findings are consistent with literature pointing to a role of midfrontal theta as a source of cognitive control signals originating from pre-frontal cortex (*Cohen and Frank, 2009*; *van Driel et al., 2012*) and alpha in posterior cortex reflecting

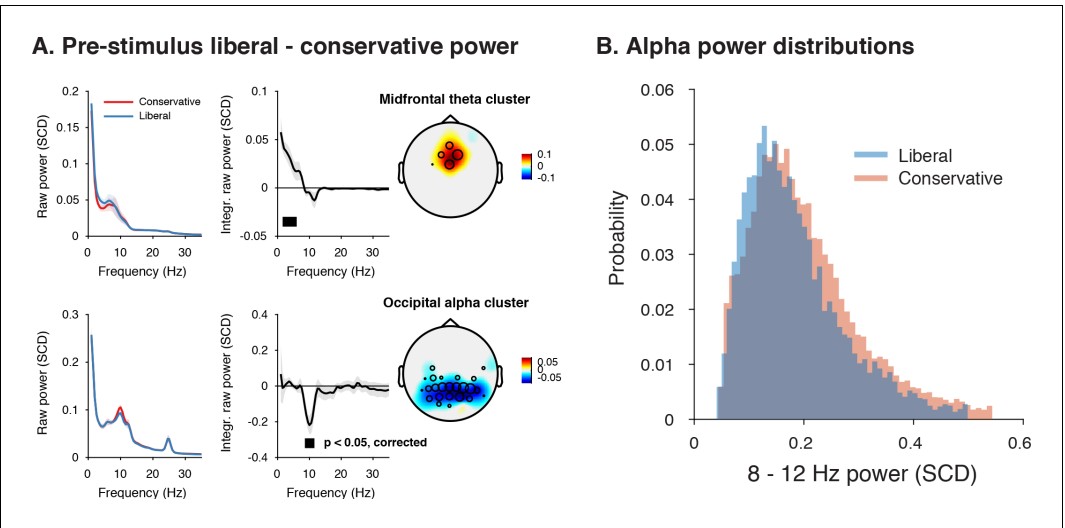

**Figure 5.** Adopting a liberal decision bias is reflected in increased midfrontal theta and suppressed pre-stimulus alpha power. (**A**) Significant clusters of power modulation between liberal and conservative in a pre-stimulus window between −1 and 0 s before trial onset. When performing a cluster-based permutation test over all frequencies (1–35 Hz) and electrodes, two significant clusters emerged: theta (2–6 Hz, top), and alpha (8–12 Hz, bottom). Left panels: raw power spectra of pre-stimulus neural activity for conservative and liberal separately in the significant clusters (for illustration purposes), Middle panels: Liberal – conservative raw power spectrum. Black horizontal bar indicates statistically significant frequency range (p<0.05, cluster-corrected for multiple comparisons, two-sided). Right panels: Corresponding liberal – conservative scalp topographic maps of the pre-stimulus raw power difference between conditions for EEG theta power (2–6 Hz) and alpha power (8–12 Hz). Plotting conventions as in *Figure 3*. Error bars, SEM across participants (N = 15). (**B**) Probability density distributions of single trial alpha power values for both conditions, averaged across participants.
DOI: https://doi.org/10.7554/eLife.37321.011

spontaneous trial-to-trial fluctuations in decision bias (*Iemi et al., 2017*). The fact that these pre-stimulus effects occur as a function of our experimental manipulation suggests that they are a hallmark of strategic bias adjustment, rather than a mere correlate of spontaneous shifts in decision bias. Importantly, this finding implies that humans are able to actively control pre-stimulus alpha power in visual cortex (possibly through top-down signals from frontal cortex), plausibly acting to bias sensory evidence accumulation toward the response alternative that maximizes reward.

## Pre-stimulus alpha power is linked to cortical gamma responses

Next, we asked how suppression of pre-stimulus alpha activity might bias the process of sensory evidence accumulation. One possibility is that alpha suppression influences evidence accumulation by modulating the susceptibility of visual cortex to sensory stimulation, a phenomenon termed 'neural excitability' (*Iemi et al., 2017*; *Jensen and Mazaheri, 2010*). We explored this possibility using a theoretical response gain model formulated by *Rajagovindan and Ding (2011)*. This model postulates that the relationship between the total synaptic input that a neuronal ensemble receives and the total output activity it produces is characterized by a sigmoidal function (*Figure 6A*) – a notion that is biologically plausible (*Destexhe et al., 2001*; *Freeman, 1979*). In this model, the total synaptic input into visual cortex consists of two components: (1) sensory input (i.e. due to sensory stimulation) and (2) ongoing fluctuations in endogenously generated (i.e. not sensory-related) neural activity. In our experiment, the sensory input into visual cortex can be assumed to be identical across trials, because the same sensory stimulus was presented in each trial (see *Figure 2A*). The endogenous input, in contrast, is thought to vary from trial to trial reflecting fluctuations in top-down cognitive processes such as attention. These fluctuations are assumed to be reflected in the strength of alpha power suppression, such that weaker alpha is associated with increased attention (*Figure 6B*). Given the combined constant sensory and variable endogenous input in each trial (see horizontal axis in *Figure 6A*), the strength of the output responses of visual cortex are largely determined by

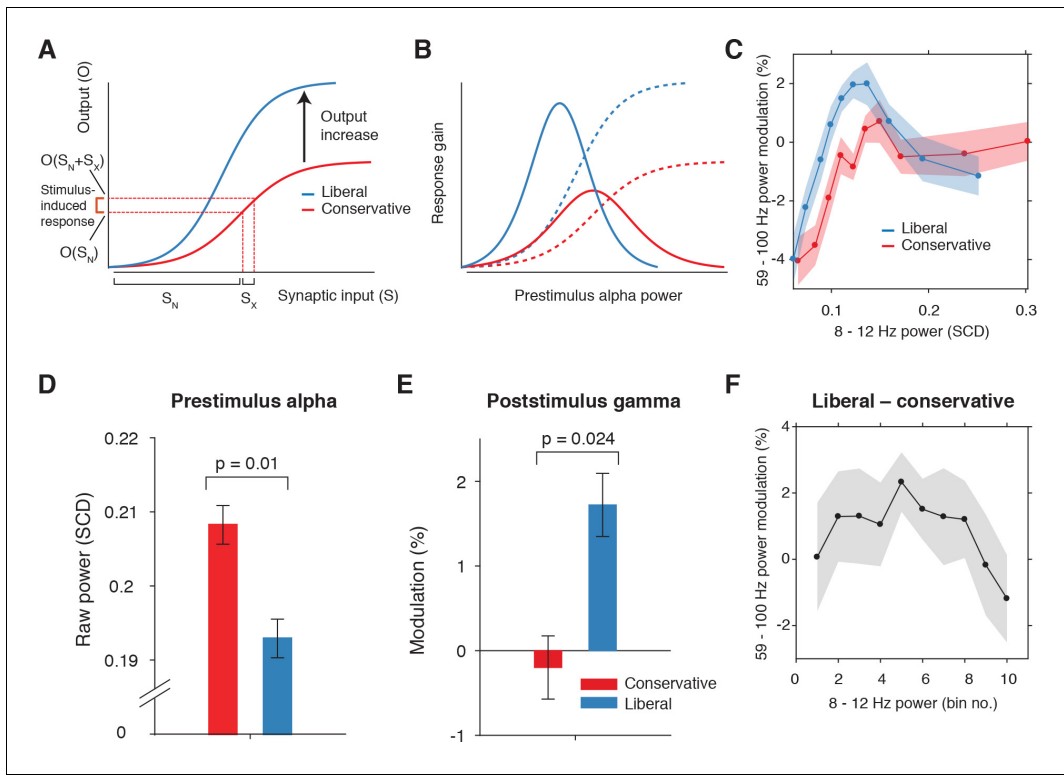

**Figure 6.** Pre-stimulus alpha power is linked to cortical gamma responses. (**A**) Theoretical response gain model describing the transformation of stimulus-induced and endogenous input activity (denoted by $S_x$ and $S_N$ respectively) to the total output activity (denoted by $O(S_x + S_N)$) in visual cortex by a sigmoidal function. Different operational alpha ranges are associated with input-output functions with different slopes due to corresponding changes in the total output. (**B**) Alpha-linked output responses (solid lines) are formalized as the first derivative (slope) of the sigmoidal functions (dotted lines), resulting in inverted-U (Gaussian) shaped relationships between alpha and gamma, involving stronger response gain in the liberal than in the conservative condition. (**C**) Corresponding empirical data showing gamma modulation (same percent signal change units as in *Figure 3*) as a function of alpha bin. The location on the x-axis of each alpha bin was taken as the median alpha of the trials assigned to each bin and averaged across subjects. (**D-F**) Model prediction tests. (**D**) Raw pre-stimulus alpha power for both conditions, averaged across subjects. (**E**) Post-stimulus gamma power modulation for both conditions averaged across the two middle alpha bins (5 and 6) in panel C. (**F**) Liberal – conservative difference between the response gain curves shown in panel C, centered on alpha bin. Error bars, within-subject SEM across participants (N = 14).

DOI: https://doi.org/10.7554/eLife.37321.012

The following source data and figure supplement are available for figure 6:

**Source data 1.** SPSS .sav file containing the data used in panels C, E, and F.
DOI: https://doi.org/10.7554/eLife.37321.014
**Figure supplement 1.** Gain model predictions and corresponding empirical data plotted as a function of pre-stimulus alpha bin number.
DOI: https://doi.org/10.7554/eLife.37321.013

the trial-to-trial variations in alpha power (see vertical axis in *Figure 6A*). Furthermore, the sigmoidal shape of the input-output function results in an effective range in the center of the function's input side which yields the strongest stimulus-induced output responses since the sigmoid curve there is steepest. Mathematically, the effect of endogenous input on stimulus-induced output responses (see marked interval in *Figure 6A*) can be expressed as the first order derivative or slope of the sigmoid in *Figure 6A*, which is referred to as the response gain by *Rajagovindan and Ding (2011)*. This derivative is plotted in *Figure 6B* (blue and red solid lines) across levels of pre-stimulus alpha power, predicting an inverted-U shaped relationship between alpha and response gain in visual cortex.

Regarding our experimental conditions, the model not only predicts that the suppression of pre-stimulus alpha observed in the liberal condition reflects a shift in the operational range of alpha (see *Figure 5B*), but also that it increases the maximum output of visual cortex (a shift from the red to the blue line in *Figure 6A*). Therefore, the difference between stimulus conditions is not modeled using a single input-output function, but necessitates an additional mechanism that changes the input-output relationship itself. The exact nature of this mechanism is not known (also see Discussion). Rajagovindan and Ding suggest that top-down mechanisms modulate ongoing prestimulus neural activity to increase the slope of the sigmoidal function, but despite the midfrontal theta activity we observed, evidence for this hypothesis is somewhat elusive. We have no means to establish directly whether this relationship exists, and can merely reflect on the fact that this change in the input-output function is necessary to capture condition-specific effects of the input-output relationship, both in the data of *Rajagovindan and Ding (2011)* and in our own data. Thus, as the operational range of alpha shifts leftwards from conservative to liberal, the upper asymptote in *Figure 6A* moves upwards such that the total maximum output activity increases. This in turn affects the inverted-U-shaped relationship between alpha and gain in visual cortex (blue line in *Figure 6B*), leading to a steeper response curve in the liberal condition resembling a Gaussian (bell-shaped) function.

To investigate sensory response gain across different alpha levels in our data, we used the post-stimulus gamma activity (see *Figure 3B*) as a proxy for alpha-linked output gain in visual cortex (*Bastos et al., 2015*; *Michalareas et al., 2016*; *Ni et al., 2016*; *Popov et al., 2017*; *van Kerkoerle et al., 2014*). We exploited the large number of trials per participant per condition (range 543 to 1391 trials) by sorting each participant's trials into ten equal-sized bins ranging from weak to strong alpha, separately for the two conditions. We then calculated the average gamma power modulation within each alpha bin and finally plotted the participant-averaged gamma across alpha bins for each condition in *Figure 6C* (see Materials and methods for details). This indeed revealed an inverted-U shaped relationship between alpha and gamma in both conditions, with a steeper curve for the liberal condition.

To assess the model's ability to explain the data, we statistically tested three predictions derived from the model. First, the model predicts overall lower average pre-stimulus alpha power for liberal than for conservative due to the shift in the operational range of alpha. This was confirmed in *Figure 6D* (p=0.01, permutation test, see also *Figure 5*). Second, the model predicts a stronger gamma response for liberal than for conservative around the peak of the gain curve (the center of the effective alpha range, see *Figure 6B*), which we indeed observed (p=0.024, permutation test on the average of the middle two alpha bins) (*Figure 6E*). Finally, the model predicts that the difference between the gain curves (when they are aligned over their effective ranges on the x-axis using alpha bin number, as shown in *Figure 6—figure supplement 1A*) also resembles a Gaussian curve (*Figure 6—figure supplement 1B*). Consistent with this prediction, we observed an interaction effect between condition (liberal, conservative) and bin number (1-10) using a standard Gaussian contrast in a two-way repeated measures ANOVA (F(1,13) = 4.6, p=0.051, partial $\eta^2$ = 0.26). *Figure 6F* illustrates this finding by showing the difference between the two curves in *Figure 6C* as a function of alpha bin number (see *Figure 6—figure supplement 1C* for the curves of both conditions as a function of alpha bin number). Subsequent separate tests for each condition indeed confirmed a significant U-shaped relationship between alpha and gamma in the liberal condition with a large effect size (F(1,13) = 7.7, p=0.016, partial $\eta^2$ = 0.37), but no significant effect in the conservative condition with only a small effect size (F(1,13) = 1.7, p=0.22, partial $\eta^2$ = 0.12), using one-way repeated measures ANOVA's with alpha bin (Gaussian contrast) as the factor of interest.

Taken together, these findings suggest that the alpha suppression observed in the liberal compared to the conservative condition boosted stimulus-induced activity, which in turn might have indiscriminately biased sensory evidence accumulation toward the target-present decision boundary. In the next section, we investigate a direct link between drift bias and stimulus-induced activity as measured through gamma.

## Visual cortical gamma activity predicts strength of evidence accumulation bias

The findings presented so far suggest that behaviorally, a liberal decision bias shifts evidence accumulation toward target-present responses (drift bias in the DDM), while neurally it suppresses pre-

stimulus alpha and enhances poststimulus gamma responses. In a final analysis, we asked whether alpha-binned gamma modulation is directly related to a stronger drift bias. To this end, we again applied the drift bias DDM to the behavioral data of each participant, while freeing the drift bias parameter not only for the two conditions, but also for the 10 alpha bins for which we calculated gamma modulation (see *Figure 6C*). We directly tested the correspondence between DDM drift bias and gamma modulation using repeated measures correlation (*Bakdash and Marusich, 2017*), which takes all repeated observations across participants into account while controlling for non-independence of observations collected within each participant (see Materials and methods for details). Gamma modulation was indeed correlated with drift bias in both conditions (liberal, r(125) = 0.49, p=5e-09; conservative, r(125) = 0.38, p=9e-06) (*Figure 7*). We tested the robustness of these correlations by excluding the data points that contributed most to the correlations (as determined with Cook's distance) and obtained qualitatively similar results, indicating these correlations were not driven by outliers (*Figure 7*, see Materials and methods for details). To rule out that starting point could explain this correlation, we repeated this analysis while controlling for the starting point of evidence accumulation estimated per alpha bin within the starting point model. To this end, we regressed both bias parameters on gamma. Crucially, we found that in both conditions starting point bias did not uniquely predict gamma when controlling for drift bias (liberal: F(1,124) = 5.8, p=0.017 for drift bias, F(1,124) = 0.3, p=0.61 for starting point; conservative: F(1,124) = 8.7, p=0.004 for drift bias, F(1,124) = 0.4, p=0.53 for starting point. This finding suggests that the drift bias model outperforms the starting point model when correlated to gamma power. As a final control, we also performed this analysis for the SSVEP (23–27 Hz) power modulation (see *Figure 3B*, bottom) and found a similar inverted-U shaped relationship between alpha and the SSVEP for both conditions (*Figure 7—figure supplement 1A*), but no correlation with drift bias (liberal, r(125) = 0.11, p=0.72, conservative, r(125) = 0.22, p=0.47) (*Figure 7—figure supplement 1B*) or with starting point (liberal, r(125) = 0.08, p=0.02, conservative, r(125) = 0.22, p=0.95). This suggests that the SSVEP is similarly

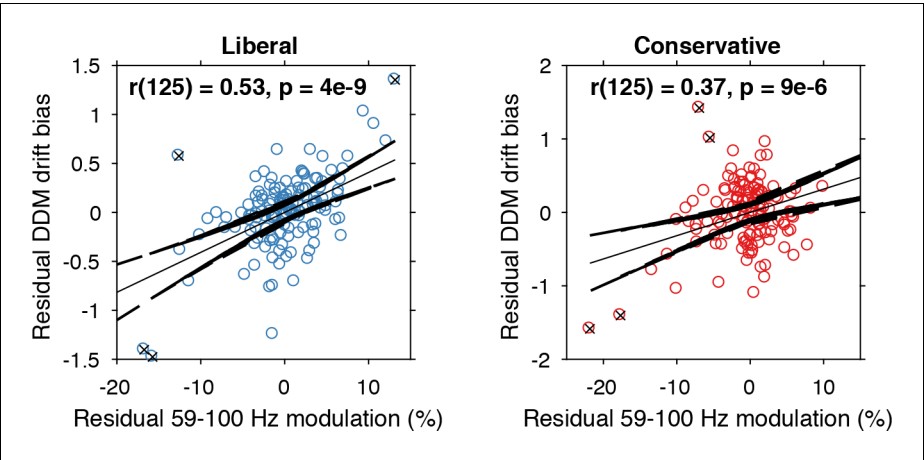

**Figure 7.** Alpha-binned gamma modulation correlates with evidence accumulation bias. Repeated measures correlation between gamma modulation and drift bias for the two conditions. Each circle represents a participant's gamma modulation within one alpha bin. Drift bias and gamma modulation scalars were residualized by removing the average within each participant and condition, thereby removing the specific range in which the participants values operated. Crosses indicate data points that were most influential for the correlation, identified using Cook's distance. Correlations remained qualitatively unchanged when these data points were excluded (liberal, r(120) = 0.46, p=8e-07; conservative, r(121) = 0.27, p=0.0009) Error bars, 95% confidence intervals after averaging across participants.

DOI: https://doi.org/10.7554/eLife.37321.015

The following source data and figure supplement are available for figure 7:

**Source data 1.** MATLAB .mat file containing the data used.
DOI: https://doi.org/10.7554/eLife.37321.017
**Figure supplement 1.** Alpha-binned post-stimulus SSVEP modulation.
DOI: https://doi.org/10.7554/eLife.37321.016

coupled to alpha as the stimulus-induced gamma, but is less affected by the experimental conditions and not predictive of decision bias shifts. Taken together, these results suggest that alpha-binned gamma modulation underlies biased sensory evidence accumulation.

Finally, we asked to what extent the enhanced tonic midfrontal theta may have mediated the relationship between alpha-binned gamma and drift bias. To answer this question, we entered drift bias in a two-way repeated measures ANOVA with factors theta and gamma power (all variables alpha-binned), but found no evidence for mediation of the gamma-drift bias relationship by midfrontal theta (liberal, $F_{(1,13)} = 1.3$, p=0.25; conservative, $F_{(1,13)} = 0.003$, p=0.95). At the same time, the gamma-drift bias relationship was qualitatively unchanged when controlling for theta (liberal, $F_{(1,13)} = 48.4$, p<0.001; conservative, $F_{(1,13)} = 19.3$, p<0.001). Thus, the enhanced midfrontal theta in the liberal condition plausibly reflects a top-down, attention-related signal indicating the need for cognitive control to avoid missing targets, but its amplitude seemed not directly linked to enhanced sensory evidence accumulation, as found for gamma. This latter finding suggests that the enhanced theta in the liberal condition served as an alarm signal indicating the need for a shift in response strategy, without specifying exactly how this shift was to be implemented (*Cavanagh and Frank, 2014*).

## Discussion

Traditionally, decision bias has been conceptualized in SDT as a criterion threshold that is positioned at an arbitrary location between noise and signal-embedded-in-noise distributions of sensory evidence strengths. The ability to strategically shift decision bias in order to flexibly adapt to stimulus-response reward contingencies in the environment presumably increases chances of survival, but to date such strategic bias shifts as well as their neural underpinnings have not been demonstrated. Here, we compared two versions of the drift diffusion model to show that an experimentally induced bias shift affects the process of sensory evidence accumulation itself, rather than shifting a threshold entity as SDT implies. Moreover, we reveal the neural signature of drift bias by showing that an experimentally induced liberal decision bias is accompanied by changes in midfrontal theta and posterior alpha suppression, resulting in enhanced gamma activity by increased response gain.

Although previous studies have shown correlations between suppression of pre-stimulus alpha (8—12 Hz) power and a liberal decision bias during spontaneous fluctuations in alpha activity (*Iemi et al., 2017*; *Limbach and Corballis, 2016*), these studies have not established the effect of experimentally induced (within-subject) bias shifts. In the current study, by experimentally manipulating stimulus-response reward contingencies we show for the first time that pre-stimulus alpha can be actively modulated by a participant to achieve changes in decision bias. Further, we show that alpha suppression in turn modulates gamma activity, in part by increasing the gain of cortical responses. Critically, gamma activity accurately predicts the strength of the drift bias parameter in the DDM drift bias model, thereby providing a direct link between our behavioral and neural findings. Together, these findings show that humans are able to actively implement decision biases by flexibly adapting neural excitability to strategically shift sensory evidence accumulation toward one of two decision bounds.

Based on our results, we propose that decision biases are implemented by flexibly adjusting neural excitability in visual cortex. *Figure 8* summarizes this proposed mechanism graphically by visualizing a hypothetical transition in neural excitability following a strategic liberal bias shift, as reflected in visual cortical alpha suppression (left panel). This increased excitability translates into stronger gamma-band responses following stimulus onset (right panel, top). These increased gamma responses finally bias evidence accumulation toward the target-present decision boundary during a liberal state, resulting in more target-present responses, whereas target-absent responses are decimated (blue RT distributions; right panel, bottom). Our experimental manipulation of decision bias in different blocks of trials suggests that decision makers are able to control this biased evidence accumulation mechanism willfully by adjusting alpha in visual cortex.

A neural mechanism that could underlie bias-related alpha suppression may be under control of the catecholaminergic neuromodulatory systems, consisting of the noradrenaline-releasing locus coeruleus (LC) and dopamine systems (*Aston-Jones and Cohen, 2005*). These systems are able to modulate the level of arousal and neural gain, and show tight links with pupil responses (*de Gee et al., 2017*; *de Gee et al., 2014*; *Joshi et al., 2016*; *Kloosterman et al., 2015b*; *McGinley et al.,*

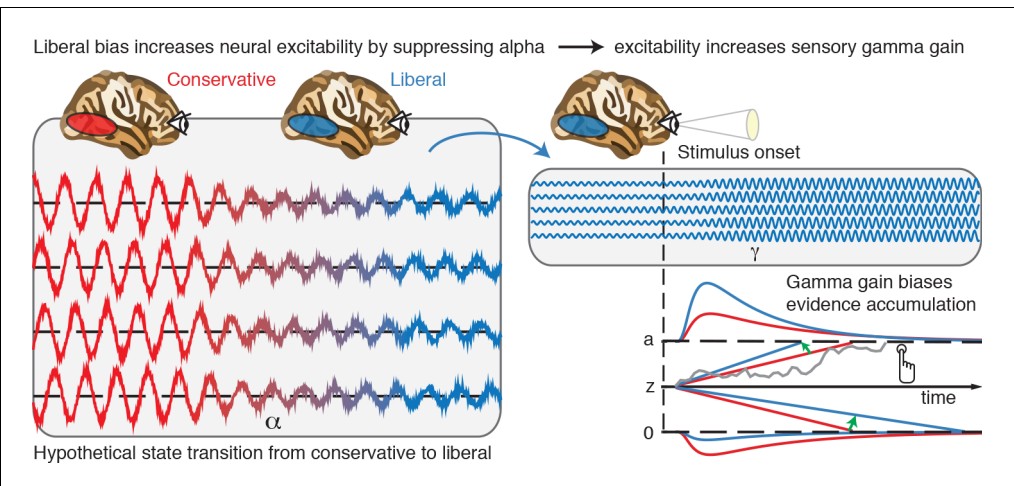

**Figure 8.** Illustrative graphical depiction of the excitability state transition from conservative to liberal, and subsequent biased evidence accumulation under a liberal bias. The left panel shows the transition from a conservative to a liberal condition block. The experimental induction of a liberal decision bias causes alpha suppression in visual cortex, which increases neural excitability. The right top panel shows increased gamma gain for incoming sensory evidence under conditions of high excitability. The right bottom panel shows how increased gamma-gain causes a bias in the drift rate, resulting in more 'target present' responses than in the conservative state.

DOI: https://doi.org/10.7554/eLife.37321.018

*2015*). Accordingly, pre-stimulus alpha power suppression has also recently been linked to pupil dilation (*Meindertsma et al., 2017*). From this perspective, our results may help to reconcile previous studies showing relationships between a liberal bias, suppression of spontaneous alpha power and increased pupil size. Consistent with this, a recent monkey study observed increased neural activity during a liberal bias in the superior colliculus (*Crapse et al., 2018*), a mid-brain structure tightly interconnected with the LC (*Joshi et al., 2016*). Taken together, a more liberal within-subject bias shift (following experimental instruction and/or reward) might activate neuromodulatory systems that subsequently increase cortical excitability and enhance sensory responses for both stimulus and 'noise' signals in visual cortex, thereby increasing a person's propensity for target-present responses (*Iemi et al., 2017*).

We note that although the gain model is consistent with our data as well as the data on which the model was conceived (see *Rajagovindan and Ding, 2011*), we do not provide a plausible mechanism that could bring about the steepening in the U-curved function observed in *Figure 6C F*. Although Rajagovindan and Ding report a simulation in their paper suggesting that increased excitability could indeed cause increased gain, this shift could in principle either be caused by the alpha suppression itself, by the same signal that causes alpha suppression, or it could originate from an additional top-down signal from frontal brain regions. Our analysis of pre-stimulus signals indeed shows preliminary evidence for such a top-down signal, but how exactly the gain enhancement arises remains an open question that could be addressed in future research.

Whereas we report a unique link between alpha-linked gamma modulation and decision bias through the gain model, several previous studies have reported a link between alpha and objective performance instead of bias, particularly in the phase of alpha oscillations (*Busch et al., 2009*; *Mathewson et al., 2009*). Our findings can be reconciled with those by considering that detection sensitivity in many previous studies was often quantified in terms of raw stimulus detection rates, which do not dissociate objective sensitivity from response bias (see *Figure 2B*) (*Green and Swets, 1966*). Indeed, our findings are in line with recently reported links between decision bias and spontaneous fluctuations in excitability (*Iemi et al., 2017*; *Iemi and Busch, 2018*; *Limbach and Corballis, 2016*), suggesting an active role of neural excitability in decision bias. Relatedly, one could ask whether the observed change in cortical excitability may reflect a change in target detection sensitivity (drift rate) rather than an intentional bias shift. This is unlikely because that would predict effects

opposite to those we observed. We found increased excitability in the liberal condition compared to the conservative condition; if this were related to improved detection performance, one would predict higher sensitivity in the liberal condition, while we found higher sensitivity in the conservative condition (compare drift rate to drift bias in both conditions in *Figure 2C*). This finding convincingly ties cortical excitability in our paradigm to decision bias, as opposed to detection sensitivity. Convergently, other studies also report a link between pre-stimulus low-frequency EEG activity and subjective perception, but not objective task performance (*Benwell et al., 2017*; *Iemi and Busch, 2018*).

In summary, our results suggest that stimulus-induced responses are boosted during a liberal decision bias due to increased cortical excitability, in line with recent work linking alpha power suppression to response gain (*Peterson and Voytek, 2017*). Future studies can now establish whether this same mechanism is at play in other subjective aspects of decision-making, such as confidence and meta-cognition (*Fleming et al., 2018*; *Samaha et al., 2017*) as well as in a dynamically changing environment (*Norton et al., 2017*). Explicit manipulation of cortical response gain during a bias manipulation (by pharmacological manipulation of the noradrenergic LC-NE system; (*Servan-Schreiber et al., 1990*)) or by enhancing occipital alpha power using transcranial brain stimulation (*Zaehle et al., 2010*) could further establish the underlying neural mechanisms involved in decision bias.

In the end, although one may be unaware, every decision we make is influenced by biases that operate on one's noisy evidence accumulation process. Understanding how these biases affect our decisions is crucial to enable us to control or invoke them adaptively (*Pleskac et al., 2017*). Pinpointing the neural mechanisms underlying bias in the current elementary perceptual task may foster future understanding of how more abstract and high-level decisions are modulated by decision bias (*Tversky and Kahneman, 1974*).

## Data and code sharing

The data analyzed in this study are publicly available on Figshare (*Kloosterman et al., 2018*). Analysis scripts are publicly available on Github (*Kloosterman, 2018*; copy archived at https://github.com/elifesciences-publications/critEEG).

# Materials and methods

### Key resources table

| Reagent type (species) or resource | Designation | Source or reference | Identifiers | Additional information |
|---|---|---|---|---|
| Biological sample (Humans) | Participants | This paper | | See Participants section in Materials and methods |
| Software, algorithm | MATLAB | Mathworks | MATLAB_R2016b, RRID:SCR_001622 | |
| Software, algorithm | Presentation | NeuroBS | Presentation_v9.9, RRID:SCR_002521 | |
| Software, algorithm | Custom analysis code | *Kloosterman, 2018* | https://github.com/nkloost1/critEEG | |
| Other | EEG data experimental task | *Kloosterman et al., 2018* | https://doi.org/10.6084/m9.figshare.6142940 | |

## Participants

Sixteen participants (eight females, mean age 24.1 years,±1.64) took part in the experiment, either for financial compensation (EUR 10, - per hour) or in partial fulfillment of first year psychology course requirements. Each participant completed three experimental sessions on different days, each session lasting ca. 2 hr, including preparation and breaks. One participant completed only two sessions, yielding a total number of sessions across subjects of 47. Due to technical issues, for one session only data for the liberal condition was available. One participant was an author. All participants had normal or corrected-to-normal vision and were right handed. Participants provided written informed

consent before the start of the experiment. All procedures were approved by the ethics committee of the University of Amsterdam.

Regarding sample size, our experiment consisted of 16 biological replications (participants) and either three (fifteen participants) or two (one participant) technical replications (i.e. experimental sessions). The sample size was determined based on two criteria: 1) obtaining large amounts of data per participant (thousands of trials), which is necessary to perform robust drift diffusion modelling of choice behavior and obtain reliable EEG spectral power estimates for each of the ten bins of trials that were created within participants, and 2) obtaining data from a sufficient number of participants to leverage across-subject variability in correlational analyses. Thus, we emphasized obtaining many data points per participant relative to obtaining many participants, while still preserving the ability to perform correlations across participants.

All participants were included in the signal-detection-theoretical and drift diffusion modeling analyses. One participant was excluded from the EEG analysis due to excessive noise (EEG power spectrum opposite of 1/frequency). One further participant was excluded from the analyses that included condition-specific gamma because the liberal–conservative difference in gamma in this participant was >3 standard deviations away from the other participants.

## Stimuli

Stimuli consisted of a continuous semi-random rapid serial visual presentation (rsvp) of full screen texture patterns. The texture patterns consisted of line elements approx. 0.07° thick and 0.4° long in visual angle. Each texture in the rsvp was presented for 40 ms (i.e. stimulation frequency 25 Hz), and was oriented in one of four possible directions: 0°, 45°, 90° or 135°. Participants were instructed to fixate on a red dot in the center of the screen. At random inter trial intervals (ITI's) sampled from a uniform distribution (ITI range 0.3–2.2 s), the rsvp contained a fixed sequence of 25 texture patterns, which in total lasted one second. This fixed sequence consisted of four stimuli preceding a (non-)target stimulus (orientations of 45°, 90°, 0°, 90° respectively) and twenty stimuli following the (non)-target (orientations of 0°, 90°, 0°, 90°, 0°, 45°, 0°, 135°, 90°, 45°, 0°, 135°, 0°, 45°, 90°, 45°, 90°, 135°, 0°, 135° respectively) (see *Figure 2A*). The fifth texture pattern within the sequence (occurring from 0.16 s after sequence onset) was either a target or a nontarget stimulus. Nontargets consisted of either a 45° or a 135° homogenous texture, whereas targets contained a central orientation-defined square of 2.42° visual angle, thereby consisting of both a 45° and a 135° texture. 50% of all targets consisted of a 45° square and 50% of a 135° square. Of all trials, 75% contained a target and 25% a nontarget. Target and nontarget trials were presented in random order. To avoid specific influences on target stimulus visibility due to presentation of similarly or orthogonally oriented texture patterns temporally close in the cascade, no 45° and 135° oriented stimuli were presented directly before or after presentation of the target stimulus. All stimuli had an isoluminance of 72.2 cd/m². Stimuli were created using MATLAB (The Mathworks, Inc, Natick, MA, USA; RRID:SCR_001622) and presented using Presentation version 9.9 (Neurobehavioral systems, Inc, Albany, CA, USA; RRID:SCR_002521).

## Experimental design

The participants' task was to detect and actively report targets by pressing a button using their right hand. Targets occasionally went unreported, presumably due to constant forward and backward masking by the continuous cascade of stimuli and unpredictability of target timing (*Fahrenfort et al., 2007*). The onset of the fixed order of texture patterns preceding and following (non-)target stimuli was neither signaled nor apparent.

At the beginning of the experiment, participants were informed they could earn a total bonus of EUR 30, -, on top of their regular pay of EUR 10, - per hour or course credit. In two separate conditions within each session of testing, we encouraged participants to use either a conservative or a liberal bias for reporting targets using both aversive sounds as well as reducing their bonus after errors. In the conservative condition, participants were instructed to only press the button when they were relatively sure they had seen the target. The instruction on screen before block onset read as follows: 'Try to detect as many targets as possible. Only press when you are relatively sure you just saw a target.' To maximize effectiveness of this instruction, participants were told the bonus would be diminished by 10 cents after a false alarm. During the experiment, a loud aversive sound was played after a false alarm to inform the participant about an error. During the liberal condition,

participants were instructed to miss as few targets as possible. The instruction on screen before block onset read as follows: 'Try to detect as many targets as possible. If you sometimes press when there was nothing this is not so bad'. In this condition, the loud aversive sound was played twice in close succession whenever they failed to report a target, and three cents were subsequently deducted from their bonus. The difference in auditory feedback between both conditions was included to inform the participant about the type of error (miss or false alarm), in order to facilitate the desired bias in both conditions. After every block, the participant's score (number of missed targets in the liberal condition and number of false alarms in the conservative condition) was displayed on the screen, as well as the remainder of the bonus. After completing the last session of the experiment, every participant was paid the full bonus as required by the ethical committee.

Participants performed six blocks per session lasting ca. nine minutes each. During a block, participants continuously monitored the screen and were free to respond by button press whenever they thought they saw a target. Each block contained 240 trials, of which 180 target and 60 nontarget trials. The task instruction was presented on the screen before the block started. The condition of the first block of a session was counterbalanced across participants. Prior to EEG recording in the first session, participants performed a 10-min practice run of both conditions, in which visual feedback directly after a miss (liberal condition) or false alarm (conservative) informed participants about their mistake, allowing them to adjust their decision bias accordingly. There were short breaks between blocks, in which participants indicated when they were ready to begin the next block.

## Behavioral analysis

We calculated each participant's criterion c (*Green and Swets, 1966*) across the trials in each condition as follows:

$$c = -\frac{1}{2} \left[ Z(\textit{Hit-rate}) + Z(\textit{FA-rate}) \right]$$

where hit-rate is the proportion target-present responses of all target-present trials, false alarm (FA)-rate is the proportion target-present responses of all target-absent trials, and Z(...) is the inverse standard normal distribution. Furthermore, we calculated objective sensitivity measure d' using:

$$d' = Z(\textit{Hit-rate}) - Z(\textit{FA-rate})$$

as well as by subtracting hit and false alarm rates. Reaction times (RTs) were measured as the duration between target onset and button press.

## Drift diffusion modeling of choice behavior

In order to be detected, the 40 ms-duration figure-ground targets used in our study undergo a process in visual cortex called figure-ground segregation. This process has been well characterized in man and monkey (*Fahrenfort et al., 2008*; *Lamme, 1995*; *Lamme et al., 2002*; *Supèr et al., 2003*), and results from recurrent processing to extract the surface region in visual cortex. Figure-ground segregation is known to extend far beyond the mere presentation time of the stimulus, thus providing a plausible neural basis for the evidence accumulation process. Further, a central assumption of the drift diffusion model is that the process of evidence accumulation is gradual, independent of whether sensory input is momentary. Indeed, the DDM was initially developed to explain reaction time distributions during memory retrieval, in which evidence accumulation must occur through retrieval of a memory trace within the brain, in the complete absence of external stimulus at the time of the decision (*Ratcliff, 1978*). Our observed RT distributions show the typical features that occur across many different types of decision and memory tasks, which the DDM is well able to capture, including a sharp leading edge and a long tail of the distributions (see *Figure 2—figure supplement 2*). The success of the DDM in fitting these data is consistent with previous work (e.g. *Ratcliff, 2006*) and might reflect the fact that observers modulate the underlying components of the decision process also when they do not control the stimulus duration (*Kiani et al., 2008*).

We fitted the drift diffusion model to our behavioral data for each subject individually, and separately for the liberal and conservative conditions. We fitted the model using a *G* square method based on quantile RT's (RT cutoff, 200 ms, for details, see *Ratcliff et al., 2018*), using custom code (*de Gee et al., 2018*) that was contributed to the HDDM 0.6.1 package (*Wiecki et al.,*

*2013*). The RT distributions for target-present responses were represented by the 0.1, 0.3, 0.5, 0.7 and 0.9 quantiles, and, along with the associated response proportions, contributed to *G* square. In addition, a single bin containing the number of target-absent responses contributed to *G* square. Each model fit was run six times, after which the best fitting run was kept. Fitting the model to RT distributions for target-present and target-absent choices (termed 'stimulus coding' in *Wiecki et al., 2013*), as opposed to the more common fits of correct and incorrect choice RT's (termed 'accuracy coding' in *Wiecki et al., 2013*), allowed us to estimate parameters that could have induced biases in subjects' behavior.

Parameter recovery simulations showed that letting both the starting point of the accumulation process and drift bias (an evidence-independent constant added to the drift toward one or the other bound) free to vary with experimental condition is problematic for data with no explicit target-absent responses (data not shown). Thus, to test whether shifts in drift bias or starting point underlie bias we fitted three separate models. In the first model ('fixed model'), we allowed only the following parameters to vary between the liberal and conservative condition: (i) the mean drift rate across trials; (ii) the separation between both decision bounds (i.e., response caution); and (iii) the non-decision time (sum of the latencies for sensory encoding and motor execution of the choice). Additionally, the bias parameters starting point and drift bias were fixed for the experimental conditions. The second model ('starting point model') was the same as the fixed model, except that we let the starting point of the accumulation process vary with experimental condition, whereas the drift bias was kept fixed for both conditions. The third model ('drift bias model') was the same as the fixed model, except that we let the drift bias vary with experimental condition, while the starting point was kept fixed for both conditions. We used Bayesian Information Criterion (BIC) to select the model which provided the best fit to the data (*Neath and Cavanaugh, 2012*). The BIC compares models based on their maximized log-likelihood value, while penalizing for the number of parameters.

## Distinguishing DDM drift bias and drift rate

In our task, only target-present responses were coupled to a behavioral response (button-press), so we could measure reaction times only for these responses, whereas reaction times for target-absent responses remained implicit. Thus, in our fitting procedure, the RT distributions for target-present responses were represented by the 0.1, 0.3, 0.5, 0.7 and 0.9 quantiles, and, along with the associated response proportions, contributed to G square. In addition, a single bin containing the number of target-absent responses contributed to G square. It has been shown that such a diffusion model with an implicit (no response) boundary can be fit to data with almost the same accuracy as fitting the two-choice model to two-choice data (*Ratcliff et al., 2018*). In a diffusion model with an implicit (no response) boundary, both an increase in drift rate and drift criterion would predict faster target-present responses. However, the key distinction is that an increase in drift additionally predicts more correct responses (for both target-present and target-absent responses), and an increase in drift criterion shifts the relative fraction of target-present and target-absent responses (decision bias). Because a single bin containing the number of target-absent responses contributed to G square, our fitting procedure can distinguish between decision bias versus drift rate.

## EEG recording

Continuous EEG data were recorded at 256 Hz using a 48-channel BioSemi Active-Two system (BioSemi, Amsterdam, the Netherlands), connected to a standard EEG cap according to the international 10–20 system. Electrooculography (EOG) was recorded using two electrodes at the outer canthi of the left and right eyes and two electrodes placed above and below the right eye. Horizontal and vertical EOG electrodes were referenced against each other, two for horizontal and two for vertical eye movements (blinks). We used the Fieldtrip toolbox (*Oostenveld et al., 2011*) and custom software (*Kloosterman et al., 2018*) in MATLAB R2016b (The Mathworks Inc, Natick, MA, USA; RRID:SCR_001622) to process the data (see below). Data were re-referenced to the average voltage of two electrodes attached to the earlobes.

## Trial extraction and preprocessing

We extracted trials of variable duration from 1 s before target sequence onset until 1.25 after button press for trials that included a button press (hits and false alarms), and until 1.25 s after stimulus

onset for trials without a button press (misses and correct rejects). The following constraints were used to classify (non-)targets as detected (hits and false alarms), while avoiding the occurrence of button presses in close succession to target reports and button presses occurring outside of trials: 1) A trial was marked as detected if a response occurred within 0.84 s after target onset; 2) when the onset of the next target stimulus sequence started before trial end, the trial was terminated at the next trial's onset; 3) when a button press occurred in the 1.5 s before trial onset, the trial was extracted from 1.5 s after this button press; 4) when a button press occurred between 0.5 s before until 0.2 s after sequence onset, the trial was discarded. See *Kloosterman et al., 2015a* and *Meindertsma et al. (2017)* for similar trial extraction procedures. After trial extraction, channel time courses were linearly detrended and the mean of every channel was removed per trial.

## Artifact rejection

Trials containing muscle artifacts were rejected from further analysis using a standard semi-automatic preprocessing method in Fieldtrip. This procedure consists of bandpass-filtering the trials of a condition block in the 110–125 Hz frequency range, which typically contains most of the muscle artifact activity, followed by a Z-transformation. Trials exceeding a threshold Z-score were removed completely from analysis. We used as the threshold the absolute value of the minimum Z-score within the block,+1. To remove eye blink artifacts from the time courses, the EEG data from a complete session were transformed using independent component analysis (ICA), and components due to blinks (typically one or two) were removed from the data. In addition, to remove microsaccade-related artifacts we included two virtual channels in the ICA based on channels Fp1 and Fp2, which included transient spike potentials as identified using the saccadic artefact detection algorithm from *Hassler et al. (2011)*. This yielded a total number of channels submitted to ICA of 48 + 2 = 50. The two components loading high on these virtual electrodes (typically with a frontal topography) were also removed. Blinks and eye movements were then semi-automatically detected from the horizontal and vertical EOG (frequency range 1–15 Hz; z-value cut-off four for vertical; six for horizontal) and trials containing eye artefacts within 0.1 s around target onset were discarded. This step was done to remove trials in which the target was not seen because the eyes were closed. Finally, trials exceeding a threshold voltage range of 200 µV were discarded. To attenuate volume conduction effects and suppress any remaining microsaccade-related activity, the scalp current density (SCD) was computed using the second-order derivative (the surface Laplacian) of the EEG potential distribution (*Perrin et al., 1989*).

## ERP analysis

We computed event-related potentials in electrode C4 by low-pass filtering the time-domain data up to 8 Hz followed by averaging all trials within participant per condition.

## Spectral analysis

We used a sliding window Fourier transform (*Mitra and Pesaran, 1999*); step size, 50 ms; window size, 400 ms; frequency resolution, 2.5 Hz) to calculate time-frequency representations (spectrograms) of the EEG power for each electrode and each trial. We used a single Hann taper for the frequency range of 3–35 Hz (spectral smoothing, 4.5 Hz, bin size, 1 Hz) and the multitaper technique for the 36–100 Hz frequency range (spectral smoothing, 8 Hz; bin size, 2 Hz; five tapers). See *Kloosterman et al., 2015a* and *Meindertsma et al. (2017)* for similar settings. Finally, to investigate spectral power also <3 Hz, we ran an additional time-frequency analysis with a window size of 1 s (i.e. frequency resolution 1 Hz) centered on the time point 0.5 s before trial onset (frequency range 1–35 Hz, no spectral smoothing, bin size 0.5 Hz).

Spectrograms were aligned to the onset of the stimulus sequence containing the (non)target, and (in a separate analysis) to the button press. Power modulations during the trials were quantified as the percentage of power change at a given time point and frequency bin, relative to a baseline power value for each frequency bin (*Figure 3*). We used as a baseline the mean EEG power in the interval 0.4 to 0 s before trial onset, computed separately for each condition. If this interval was not completely present in the trial due to preceding events (see Trial extraction), this period was shortened accordingly. We normalized the data by subtracting the baseline from each time-frequency bin and dividing this difference by the baseline (x 100%). For the analysis of raw pre-stimulus power

modulations, no baseline correction was applied on the raw scalp current density values. We focused our analysis of EEG power modulations around target onsets on those electrodes that processed the visual stimulus. To this end, we averaged the power modulations or raw power across eleven occipito-parietal electrodes that showed stimulus-induced responses in the gamma-band range (59–100 Hz). See *Kloosterman et al., 2015a* and *Meindertsma et al. (2017)* for a similar procedure.

## Statistical significance testing of EEG power modulations across space, time and frequency

To determine clusters of significant modulation with respect to the pre-stimulus baseline without any a priori selection, we ran statistics across space-time-frequency bins using paired t-tests across subjects performed at each bin. Single bins were subsequently thresholded at p<0.05 and clusters of contiguous time-space-frequency bins were determined. Cluster significance was assessed using a cluster-based permutation procedure (1000 permutations). For visualization purposes, we integrated (using the matlab trapz function) power modulation in the time-frequency representations (TFR's, *Figure 3*, left panels) across the highlighted electrodes in the topographies (*Figure 3*, right panels). For the topographical scalp maps, modulation was integrated across the saturated time-frequency bins in the TFRs. To test at which frequencies raw prestimulus EEG power differed between the liberal and conservative conditions, we performed this analysis across electrodes and frequencies after taking the liberal − conservative difference at each frequency bin (*Figure 5A*) (see Statistical comparisons).

## Response gain model test

To test the predictions of the gain model, we first averaged activity in the 8–12 Hz range from 0.8 to 0.2 s before trial onset (staying half our window size from trial onset, to avoid mixing pre- and post-stimulus activity, also see *Iemi et al., 2017*), yielding a single scalar alpha power value per trial. If this interval was not completely present in the trial due to preceding events (see Trial extraction), this period was shortened accordingly. Trials in which the scalar was >3 standard deviations away from the participant's mean were excluded. We then sorted all single-trial alpha values for each participant and condition in ascending order and assigned them to ten bins of equal size, ranging from weakest to strongest alpha. Adjacent bin ranges overlapped for 50% to stabilize estimates. Then we averaged the corresponding gamma modulation of the trials belonging to each bin (consisting of the average power modulation within 59–100 Hz 0.2 to 0.6 s after trial onset, see *Figure 3*). Finally, we averaged across participants and plotted the median alpha value per bin averaged across participants against the mean gamma modulation. See *Rajagovindan and Ding (2011)* for a similar procedure. To statistically test for the existence of inverted U-shaped relationships between alpha and gamma, we performed a one-way repeated measures ANOVA on gamma modulation with factor alpha bin (10 bins) to each condition separately and a two-way repeated measures ANOVA with factors bin and condition for testing the liberal–conservative difference (*Figure 6F*). Given the model prediction of a Gaussian-shaped relationship between alpha and gamma, we constructed a Gaussian contrast using a Gaussian shape with unit standard deviation (contrast values: −1000,−991, −825, 295, 2521, 2521, 295,−825, −991,−1000, values were chosen to sum to zero). For plotting purposes (*Figure 6C-F*), we computed within-subject error bars by removing within each participant the mean across conditions from the estimates.

## Correlation between gamma modulation and drift bias

To link DDM drift bias and gamma power modulation, we re-fitted the DDM drift bias model while freeing the drift bias parameter both for each condition as well as for the ten alpha bins, while freeing the other parameters (drift rate, boundary separation, non-decision time) for each condition and fixing starting point across conditions. We then used repeated measures correlation to test whether stronger gamma was associated with stronger drift bias. Repeated measures correlation determines the common within-individual association for paired measures assessed on two or more occasions for multiple individuals by controlling for the specific range in which individuals' measurements operate, and correcting the correlation degrees of freedom for non-independence of repeated measurements obtained from each individual. Specifically, the correlation degrees of freedom were 14 participants $\times$ 10 observations – Number of participants – 1 = 140 – 14 – 1 = 125. Repeated

measures correlation tends to have greater statistical power than conventional correlation across individuals because neither averaging nor aggregation is necessary for an intra-individual research question. Please see *Bakdash and Marusich (2017)* for more information. We assessed the impact of single observations on the correlations by excluding observations exceeding five times the average Cook's distance of all values within each condition (five observations for liberal and four for conservative) and recomputing the correlations.

## Statistical comparisons

We used two-sided permutation tests (10,000 permutations) (*Efron and Tibshirani, 1998*) to test the significance of behavioral effects and the model fits. Permutation tests yield p=0 if the observed value falls outside the range of the null distribution. In these cases, p<0.0001 is reported in the manuscript. The standard deviation (s.d.) is reported as a measure of spread along with all participant-averaged results reported in the text. To quantify power modulations after (non-)target onset, we tested the overall power modulation for significant deviations from zero. For these tests, we used a cluster-based permutation procedure to correct for multiple comparisons (*Maris and Oostenveld, 2007*). For time-frequency representations along with spatial topographies of power modulation, this procedure was performed across all time-frequency bins and electrodes; for frequency spectra across all electrodes and frequencies; for power and ERP time courses, across all time bins. To test the existence of inverted-U shaped relationships between gamma and alpha bins, we conducted repeated measures ANOVA's and Gaussian shaped contrasts (see section Response gain model test for details) using SPSS 23 (IBM, Inc). We used multiple regression to assess whether starting point could account for the correlation between gamma and drift bias. We used Pearson correlation to test the link between parameter estimates of the DDM and SDT frameworks and repeated measures correlation to test the link between gamma power and drift bias (see previous section).

## Acknowledgements

The authors thank Timothy J Pleskac for discussion.

## Additional information

### Funding

| Funder | Grant reference number | Author |
|---|---|---|
| Max-Planck-Gesellschaft | Open-access funding | Niels A Kloosterman<br>Markus Werkle-Bergner<br>Ulman Lindenberger<br>Douglas D Garrett |
| Deutsche Forschungsgemeinschaft | Emmy Noether Programme grant | Niels A Kloosterman<br>Douglas D Garrett |
| Max Planck UCL Centre for Computational Psychiatry and Ageing Research | | Niels A Kloosterman<br>Ulman Lindenberger<br>Douglas D Garrett |
| Jacobs Foundation | Early Career Research Fellowship | Markus Werkle-Bergner |
| Deutsche Forschungsgemeinschaft | WE4296/5-1 | Markus Werkle-Bergner |

The funders had no role in study design, data collection and interpretation, or the decision to submit the work for publication.

### Author contributions

Niels A Kloosterman, Conceptualization, Data curation, Software, Formal analysis, Investigation, Visualization, Methodology, Writing—original draft, Project administration, Writing—review and editing; Jan Willem de Gee, Resources, Software, Formal analysis, Methodology, Writing—review and editing; Markus Werkle-Bergner, Conceptualization, Methodology, Writing—review and editing;

Ulman Lindenberger, Resources, Funding acquisition, Writing—review and editing; Douglas D Garrett, Resources, Formal analysis, Supervision, Funding acquisition, Investigation, Methodology, Writing—review and editing; Johannes Jacobus Fahrenfort, Conceptualization, Data curation, Software, Formal analysis, Supervision, Visualization, Methodology, Writing—original draft, Project administration, Writing—review and editing

### Author ORCIDs
Niels A Kloosterman (iD) http://orcid.org/0000-0002-1134-7996
Jan Willem de Gee (iD) https://orcid.org/0000-0002-5875-8282
Markus Werkle-Bergner (iD) http://orcid.org/0000-0002-6399-9996
Ulman Lindenberger (iD) https://orcid.org/0000-0001-8428-6453
Douglas D Garrett (iD) https://orcid.org/0000-0002-0629-7672
Johannes Jacobus Fahrenfort (iD) http://orcid.org/0000-0002-9025-3436

### Ethics
Human subjects: Participants provided written informed consent before the start of the experiment. All procedures were approved by the ethics committee of the Psychology Department of the University of Amsterdam (approval identifier: 2007-PN-69).

### Decision letter and Author response
Decision letter https://doi.org/10.7554/eLife.37321.023
Author response https://doi.org/10.7554/eLife.37321.024

## Additional files

### Supplementary files
• Transparent reporting form
DOI: https://doi.org/10.7554/eLife.37321.019

### Data availability
All data analysed during this study are publicly available, see https://doi.org/10.6084/m9.figshare.6142940.v1. Analysis scripts are publicly available on Github (https://github.com/nkloost1/critEEG; copy archived at https://github.com/elifesciences-publications/critEEG).

The following dataset was generated:

| Author(s) | Year | Dataset title | Dataset URL | Database and Identifier |
|---|---|---|---|---|
| Niels A. Kloosterman, Jan Willem de Gee, Markus Werkle-Bergner, Ulman Lindenberger, Douglas D Garrett, Johannes Jacobus Fahrenfort | 2018 | Humans strategically shift decision bias by flexibly adjusting sensory evidence accumulation in visual cortex | https://doi.org/10.6084/m9.figshare.6142940 | figshare, 10.6084/m9.figshare.6142940 |

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
