## [Decision Letter]

Thank you for submitting your article "Humans strategically shift decision bias by flexibly adjusting sensory evidence accumulation in visual cortex" for consideration by *eLife*. Your article has been reviewed by three peer reviewers, including Ole Jensen as the Reviewing Editor and Reviewer #1, and the evaluation has been overseen by David Van Essen as the Senior Editor. The following individual involved in review of your submission has agreed to reveal their identity: Eelke Spaak (Reviewer #2).

The reviewers have discussed the reviews with one another and the Reviewing Editor has drafted this decision to help you prepare a revised submission.

Summary:

The paper by Kloosterman et al. has been using a visual paradigm and EEG to investigate evidence accumulation in the visual cortex in humans. The data were fitted using a diffusion drift model. Key to the experiment was that the participants' decision criteria were manipulated using different reward contingencies inducing liberal or conservative biases. A liberal bias resulted in a suppression of the pre-stimulus alpha power and a subsequent stimulus induced gamma increase. It is stated that the alpha decrease boosted the gamma increase. These are in principle interesting findings speaking to how decision-making relates to oscillatory brain activity in the context of bias. All reviewers judge the paper of potential interest; however, several serious concerns pertaining to the analysis were raised. In particular the result summary/conclusion in the Abstract and Discussion seems too simplistic in the light of the actual findings (in particular the inverted-U of alpha in relation to gamma as explained below). Furthermore additional work is needed on the analysis. Specifically, the excitability model seems at odds with the interpretation that alpha-power regulates the drift bias as the correlation between alpha-power and gamma is different for conservative vs liberal (which should not be the case if conservative vs liberal modulates excitability through alpha). Please see below for details.

Essential revisions:

1) The authors show that pre-stimulus alpha power depends on the condition and hypothesize that this is involved in regulating the gain – somewhat plausibly supported by showing the correlation of alpha-power with gamma-power in a non-linear fashion (inverted U-shape). The issue is that the predicted difference between conservative and liberal trials in the (nonlinear) correlation between gamma- and alpha is not logically explained if alpha-power is equated to neuronal excitation. If alpha is taken as an index of neural excitation, then this is not the predicted result that emerges from Figure 5A (which shows stimulus effect against membrane excitability). Following this logic, the conservative vs. liberal condition should result in different histograms of high vs. low alpha power states, but these should not change the profile of the dependency of the stimulation effect on excitability (as indexed by alpha).

In this respect, it would be relevant to see whether alpha vs. drift bias shows a similar (nonlinear) correlation. Wouldn't one expect to observe 1) a nonlinear correlation between gamma and decision bias? 2) the correlation to be present both in the liberal and conservative condition?

This central issue needs to be addressed: if the model is correct, one would expect to see the very same correlation between alpha and gamma for conservative and liberal trials, just with the distribution of the liberal trials being shifted toward lower alpha-power and thereby more excitable. The current result in Figure 5C (and their prediction from Figure 5B), if valid, suggests an additional mechanism, for instance a dynamic (differential) top-down signal from areas like SMA or DLPFC into visual cortex that might explain the differential correlation. We suggest to show the results of the contrast liberal vs. conservative across all electrodes (ideally with an average reference) – with a specific question on whether this shows differences in power in more frontal areas. The cluster permutation method applied is usually appropriate for correcting for multiple comparisons across time, frequency and sensors.

2) We would think that the SSVEPs directly reflect excitability. Is there a reason for not performing the analysis for the SSVEPs? Such an analysis might help to clarify the point above.

3) Most studies on evidence accumulation are applying continuous stimuli (e.g. random dot kinematograms) in which information gradually is accumulated. In this study the informative target is shown for 40 ms. I take it lasts longer to accumulate information in order to make a decision? Please clarify.

4) In Figure 5 (and elsewhere) 'excitability' denotes alpha suppression. Why not just label it 'alpha suppression' or alike? While 'excitability' and 'alpha suppression' are related, one cannot equate them.

5) The participants only made yes responses. How can one then distinguish between decision bias versus drift rate? (Only the 'upper arrows' in Figure 2D are present in the data).

6) Figure 5C is essential for making the claim on the relationship between alpha and gamma power. However it is not clear from caption or Materials and methods section how this plot is produced. We take that alpha suppression is sorted in 10 bins per subject. The description of the 'neural gain analysis' (subsection “Α suppression enhances the gain of cortical γ responses”, Figure 5, and associated Materials and methods section) is unclear, which leaves us unable to fully judge its correctness. We understand that the output of a region is considered a (sigmoidal) function of total input, where total input is the sum of stimulus-related and endogenous input. Why is it such that "the isolated effect of *sensory* input […] can then be expressed as the first-order derivative of the sigmoid"? It seems to us that this derivative would be the effect of *any* input. This mistake is a symptom of the authors sometimes conflating gain (which we would equate with the slope of the output/input curve) and actual input or output. Relatedly, the authors write "stimulus-related output gain" where they actually mean "output"; i.e. it is (if we understand correctly) precisely the output that is *not* stimulus-related which is relevant here, namely the endogenous fluctuations. There is confusion between gain, input, and output in some other places in this description as well; also how these terms map onto experimental measures is a bit ambiguous. (Gain = liberal vs. conservative; input = alpha; output = gamma? This is what we understand, but at times it appears as though alpha is equated to gain instead of input.)

7) The 3-way ANOVA reported in the last paragraph of the subsection “Α suppression enhances the gain of cortical γ responses” is, we believe, not the correct way to analyze these data. The dependent variable here is gamma power, with independent variables condition (liberal/conservative) and alpha power bin (10 levels). Thus, a 2-way RM-ANOVA would be appropriate. If the authors believe the 3-way approach is indeed correct, then they should explain why this is so.

8) The approach taken for the "within-subject group regression" is unclear to us (also not explained in Materials and methods). The primary evidence that links gamma activity to DDM drift bias is, it appears, based on regressing drift bias onto gamma power across different alpha bins, where both variables are averaged within bin, across participants (Figure 6). The correct approach here would be to perform this regression per participant, and then test whether the regression coefficients are different from zero at the population level. (Or better yet, show the individual regression lines.)

9) The description of report proportions (Figure 2B) is not clearly defined. Shouldn't these sum to 1 within a condition? Additionally, it would be good to have some feeling of absolute number of responses, including those responses counting as a miss/correct rejection.

[Editors' note: further revisions were requested prior to acceptance, as described below.]

Thank you for resubmitting your work entitled "Humans strategically shift decision bias by flexibly adjusting sensory evidence accumulation in visual cortex" for further consideration at *eLife*. Your article has been reviewed by two peer reviewers, and the evaluation has been overseen by a Senior/Reviewing Editor.

The two reviewers and I have reviewed your response and the new manuscript. While all agreed that the manuscript has improved and the overall question is interesting, there was some disagreement as to the value of the work given the limitations. As such there remain some essential revisions that will be paramount to address properly in another revision, at which point we may need to solicit input from another reviewer in case there remains disagreement.

The two major strengths of the paper are the following:

1) The fact that you demonstrate fairly clearly that there is an effect of liberal vs. conservative incentive on some form of "response gain" parameter, i.e. a change in the response time distribution that apparently cannot be explained by a change in a constant alone but a gain parameter (by model fits).

2) The fact that this (behaviourally effective) conservative vs. liberal task-set has a pronounced effect on pre-stimulus occipital alpha oscillations and also on stimulus induced gamma-oscillations.

However, the following weaknesses were noted and should be addressed directly. Please note that there is no guarantee of acceptance.

1) The fact that the occipital alpha power is not clearly the decisive element here, but rather seems to be the consequence of a different signal of currently unknown (and still clearly under-investigated) origin that actually mediates the "drift bias". The idea that these dynamics are predominantly taking place within the visual system – as suggested in title and Abstract – is not sufficiently justified by evidence. This mostly amounts to a matter of emphasis – you do acknowledge that this "remains an open question" and that alpha modulation is not the whole story. Having said that, it is clear that the text (title Abstract and elsewhere) can be interpreted as trying to make such a point. So, I would suggest that a further rewrite on this point is in order, perhaps adding some of the details in the response-to-reviewers document to the manuscript (Discussion/supplement) itself.

One of the reviewers noted that you stated in your response that you updated the description of the Rajagovindan and Ding model and now account for the fact that an additional signal is required to explain these results – but the detailed explanation of this is lacking in the central description of this model in the subsection “Pre-stimulus alpha power mediates cortical gamma responses” (the second paragraph is not clear to understand and hides the fact that another external factor is needed here).

2) Given the debate about whether you have identified the critical neural mechanism/correlate of the behavioral effect, it is even more important, for the potential novelty and significance of the results, to demonstrate that your results are specific to drift bias as opposed to starting point. This distinction is theoretically important, but the main conclusion in support of drift bias account is based on model fit metrics (BIC) alone. To be more confident that this difference is meaningful it is important to provide evidence that the drift bias model empirically captures the RT distributions better than the starting point model. In theory the better BIC fit indicates the drift bias does capture RT distributions better, but sometimes models can fit better using BIC (or other such metrics) for nuisance reasons unrelated to the core theoretical distinction. You do show individual subject RT distribution fits in Figure 2—figure supplement 3 for the drift bias model which look quite reasonable, but you do not show the corresponding fits to the starting point model, and the main claim of the paper rests on the ability to distinguish these with your task design/models.

3) Relatedly, you do provide evidence that stimulus activity in gamma is amplified in the liberal condition, which is consistent with a drift bias, and also that this correlates with the extent of drift bias across subjects. What would be nice here is if you showed that this correlation was specific to the drift bias model and that it was statistically more evident than the corresponding correlations with starting point bias (i.e., you could test if the stimulus gamma activity is similarly correlated with estimated starting point bias in the alternative model). This would provide another more specific test of the evidence for the link between EEG and behavior even without identifying the top-down mechanism.

---

## [Author Response]

Essential revisions:1) The authors show that pre-stimulus alpha power depends on the condition and hypothesize that this is involved in regulating the gain – somewhat plausibly supported by showing the correlation of alpha-power with gamma-power in a non-linear fashion (inverted U-shape). The issue is that the predicted difference between conservative and liberal trials in the (nonlinear) correlation between gamma- and alpha is not logically explained if alpha-power is equated to neuronal excitation. If alpha is taken as an index of neural excitation, then this is not the predicted result that emerges from Figure 5A (which shows stimulus effect against membrane excitability). Following this logic, the conservative vs. liberal condition should result in different histograms of high vs. low alpha power states, but these should not change the profile of the dependency of the stimulation effect on excitability (as indexed by alpha).

Thank you for pointing out this important issue. In our initial manuscript submission, we presented a simplified version of the model originally described by Rajagovindan and Ding ((2011); from here on simply referred to as R&D). We agree however with the reviewers that this simplified R&D model merely predicted that alpha-band activity (our proxy of neural excitability) can boost gamma-band activity (our proxy of output activity of visual cortex) by regulating where the stimulus induced activity passes through the transfer function; the simplified R&D model did not predict a change in the slope of the hypothesized transfer function itself. Thus, we agree that our simplified R&D model did not explain our important finding that the U-shaped relationship between alpha- and gamma-band activity was steeper in the liberal compared to the conservative condition.

We now choose to present the full R&D model as described in Rajagovindan and Ding (2011). This full R&D model indeed posits that attention not only changes the neural excitability, but also the maximum total output of visual cortex. This latter effect culminates in a change in slope of the transfer function (see Figure 7A in Rajagovindan and Ding (2011)). We now present the full R&D model in Figures 5A and 5B, which predicts the increased steepening of the U-curve as a function of (attentional) condition (Figure 5B). As such, we believe that the full R&D model is in accordance with our empirical result that the U-shaped relationship between alpha- and gamma-band activity is steeper in the liberal compared to the conservative condition (Figure 5C).

Please note that although we follow R&D’s model in our manuscript, we agree that it is not necessarily the most parsimonious model, because it requires an extra mechanism on top of the inverted U-shaped effect of alpha on gamma. As the reviewers suggest, a simpler model would have been one in which there is only a single input-output relationship (such as depicted in our original Figure 5A), that does not change as a function of condition. However, such a simple model cannot explain our data. If this were the case, the shift of the range in which alpha occurs (as can be seen in Figure 4C, but also in 5C) would have to be large enough to make the distribution fall outside the range in which alpha suppression maximally drives gamma. As can be seen in Figure 5C, this is not what we see in our data. Conversely, the R&D model depicted in Figure 5B makes three predictions that are in line with our data, which we now make explicit in Figure 5D-5F: 1) overall lower alpha power for liberal than for conservative due to the shift in the effective range of alpha (Figure 5D) 2) A stronger gamma response in the peak (the center of the effective alpha range) for liberal than for conservative (Figure 5E), and 3) the difference between the alpha-gamma functions in the two conditions (when mapping the alpha-ranges in which they operate onto each other), results again in an inverted U-shape (Figure 5F and Figure 5—figure supplement 1). Please see the rationale for the related ANOVA and its outcome under point 7. Below, we further discuss the plausibility of an extra mechanism that changes the steepness of the input-output function.

In this respect, it would be relevant to see whether alpha vs. drift bias shows a similar (nonlinear) correlation.

Indeed, we observe an inverse-U shaped between alpha and drift bias for both conditions, as shown in Author response image 1. This correlation follows logically given that alpha and gamma also correlate non-linearly (see Figure 5C) and gamma and drift bias are linearly correlated (see Figure 6).

**Author response image 1. respfig1:** Inverse-U shaped relationship between pre-stimulus alpha power and drift bias. DDM drift bias parameter estimates were Z-scored within each condition to remove the large inter-individual differences in average drift bias within each condition and focus on the shape of the relationship between alpha and drift bias. Therefore, the difference in average drift bias between conditions (see Figure 2F) is not reflected in these figures.

Wouldn't one expect to observe:1) a nonlinear correlation between gamma and decision bias?

On theoretical grounds, we hypothesized that gamma (output of visual cortex) is a neural reflection of drift bias (speed of accumulation toward a decision boundary). This predicts that a linear increase in gamma is expressed in a linear increase in drift bias, hence a linear correlation. This is indeed what we observed (see Figure 6). The only non-linear relation we hypothesized to exist is between alpha and gamma, where alpha only drives gamma in an effective range (hence the inverted U-shape). This is also what we observe: a non-linear relation between alpha and gamma (Figure 5C), but a linear relationship between gamma and drift bias (see Figure 6).

2) The correlation to be present both in the liberal and conservative condition?This central issue needs to be addressed: if the model is correct, one would expect to see the very same correlation between alpha and gamma for conservative and liberal trials, just with the distribution of the liberal trials being shifted toward lower alpha-power and thereby more excitable.

Thank you for pointing this out. Indeed, we agree that the model predicts a correlation between gamma and drift bias in both conditions, albeit with lower gamma and drift bias values in the conservative than in the liberal condition. While investigating this, we identified a number of shortcomings in both the initial alpha-gamma coupling analysis and the initial correlation analysis, which we improved as follows:

- Estimating the DDM drift bias parameter separately for each of the ten alpha bins requires a sufficient number of trials per bin to build a distribution of reaction times to be fitted by the DDM. However, we realized that our initial fixed-bin-width alpha binning procedure did not guarantee this for all bins, because fewer trials were assigned to the extreme bins due to the two-tailed shape of the single trial alpha distributions (see Figure 4C). Instead, we now bin the trials using equal-sized bins, such that by design each bin has the same number of trials (within each participant and condition). Although this change in binning procedure does not qualitatively change the outcome of the alpha-gamma analysis as reported in our original submission (Figure 5C), we found that the DDM drift bias estimates are less noisy with equal-sized bins, enhancing our ability to detect an effect.

- Further, we now use the same percent signal change normalization of post-stimulus gamma power in the alpha-versus-gamma analysis (Figure 5C) as in the analysis of stimulus-related responses (Figure 3). Specifically, we take the condition-specific pre-stimulus baseline spectrum (-0.4 to 0 s), and express modulation in percent signal change (psc) with respect to this baseline (see Figures 3 and 5). These psc values now go directly into the final behavioral correlation analysis, without z-scoring the data (see Figure 6). This unified approach has both sharpened the responses observed in the different analyses, and has made the analysis trajectory throughout the paper more transparent.

- Finally, instead of the within-subject group regression, we now use a more sensitive repeated measures correlation approach. This approach and its rationale are explained under point 8 below.

Due to these improvements in the analysis, we now observe a significant effect of condition on the slope of the inverted U-curve (Figure 5), and detect robust positive correlations between gamma and drift bias in both conditions, in line with the predictions of the gain model. See Figure 6 for the results of this improved correlation analysis. Figure 5E confirms that gamma operates in a higher range in the liberal condition, in line with the observed stronger drift bias in the liberal condition.

The current result in Figure 5C (and their prediction from Figure 5B), if valid, suggests an additional mechanism, for instance a dynamic (differential) top-down signal from areas like SMA or DLPFC into visual cortex that might explain the differential correlation. We suggest to show the results of the contrast liberal vs. conservative across all electrodes (ideally with an average reference) – with a specific question on whether this shows differences in power in more frontal areas. The cluster permutation method applied is usually appropriate for correcting for multiple comparisons across time, frequency and sensors.

We agree with the reviewers that we do not uncover a plausible mechanism that could bring about the steepening in the U-curved function observed in Figure 5C. This shift could either be caused by the same signal that causes alpha suppression, by the alpha suppression itself, or it could originate from an additional top-down signal from frontal brain regions. To test whether any frontal brain region shows differences between conditions, we performed the suggested liberal–conservative contrast across space, time and frequency, using a condition-average baseline correction. In this exploratory analysis, we did not pre-select any electrodes, time or frequency bins, but instead identified clusters of space-time-frequency bins that showed significant differences between conditions. These clusters were corrected for multiple comparisons using the cluster permutation procedure in FieldTrip. See Meindertsma et al., 2018, for a similar approach. We did not find any frontal clusters in this analysis, even when using a less stringent test by omitting the required correction for multiple comparisons. We report this result in the fifth paragraph of the Discussion and have added this as a supplementary figure to the manuscript (Figure 3—figure supplement 1).

We note, however, that R&D report a simulation in their paper exploring the relationship between cortical excitability and gain. This simulation indicates that an intermediate excitability state results in a steeper sigmoid transfer function between background synaptic activity and output firing rate, whereas states of lower and higher excitability yield shallower transfer functions (see Figure 10 from Rajagovindan and Ding, 2011). This result suggests that the stronger alpha suppression in the liberal compared to the conservative condition might indeed reflect a steepened transfer function slope (enhanced gain) due to increased excitability. Although this model fits our and R&D’s empirical findings, it does by itself not explain the mechanism by which the gain is enhanced. Although we deem this a very important issue, we did not see opportunities to further characterize this mechanism beyond the findings we already present. We have now made this issue explicit, added a description of this model limitation to the Discussion section (fifth paragraph), and propose this issue as a topic for future research.

2) We would think that the SSVEPs directly reflect excitability. Is there a reason for not performing the analysis for the SSVEPs? Such an analysis might help to clarify the point above.

We agree with the reviewers that the strength of the pre-stimulus SSVEP could in principle reflect excitability, because it can be seen as a readout of the responsiveness of visual cortex to external input. Figures 4A and 4B, however, show that besides the robust alpha band suppression during the liberal condition, there is no significant pre-stimulus difference between conditions in the SSVEP frequency range, indicating that the SSVEP does not differentiate the two conditions in terms of excitability.

We also investigated the effect of pre-stimulus alpha level on the strength of the post-stimulus SSVEP modulation, and observed a similar U-shaped relationship as for gamma, suggesting that SSVEP power more closely reflects the output of visual cortex than that it reflects excitability itself. However, the Gaussian (inverted-U shaped) contrast on SSVEP across alpha bins in the two-way repeated measures ANOVA was not significant for any condition, nor was it significantly different between conditions. Finally, the alpha-binned SSVEP modulation was not significantly correlated with drift bias, as observed for gamma, although the correlations were in the same (positive) direction as for gamma. Taken together, these results suggest that the stimulus-related SSVEP shows a similar coupling to alpha as the stimulus-induced gamma, but is less affected by the experimental conditions and not predictive of criterion shifts, like gamma. We now report these findings in the Results subsection “Visual cortical gamma activity predicts strength of evidence accumulation bias” and have added them to the manuscript as a supplementary figure (Figure 6—figure supplement 1).

3) Most studies on evidence accumulation are applying continuous stimuli (e.g. random dot kinematograms) in which information gradually is accumulated. In this study the informative target is shown for 40 ms. I take it lasts longer to accumulate information in order to make a decision? Please clarify.

Thank you for raising this important issue. We have now added the following text to the subsection “Drift diffusion modeling of choice behavior”:

“In order to be detected, the 40 ms-duration figure-ground targets used in our study undergo a process in visual cortex called figure-ground segregation. […] The success of the DDM in fitting these data is consistent with previous work (e.g. Ratcliff, 2006) and might reflect the fact that observers modulate the underlying components of the decision process also when they do not control the stimulus duration (Kiani, Hanks, and Shadlen, 2008).”

4) In Figure 5 (and elsewhere) 'excitability' denotes alpha suppression. Why not just label it 'alpha suppression' or alike? While 'excitability' and 'alpha suppression' are related, one cannot equate them.

Thank you for pointing this out. We fully agree and carefully revised the manuscript to always refer to ‘alpha suppression’, and only when interpreting the results, referring to terms such as excitability.

5) The participants only made yes responses. How can one then distinguish between decision bias versus drift rate? (only the 'upper arrows' in Figure 2D are present in the data).

Thank you for raising this important question, we realize we had not sufficiently explained this in the previous version of the manuscript. We have now added the following text to the subsection “Distinguishing DDM drift bias and drift rate”:

“Distinguishing DDM drift bias and drift rateIn our task, only target-present responses were coupled to a behavioral response (button-press), so we could measure reaction times only for these responses, whereas reaction times for target-absent responses remained implicit. […] Because a single bin containing the number of target-absent responses contributed to G square, our fitting procedure can distinguish between decision bias versus drift rate.”

6) Figure 5C is essential for making the claim on the relationship between alpha and gamma power. However it is not clear from caption or Materials and methods section how this plot is produced. We take that alpha suppression is sorted in 10 bins per subject. The description of the 'neural gain analysis' (subsection “Alpha suppression enhances the gain of cortical gamma responses”, Figure 5, and associated Materials and methods section) is unclear, which leaves us unable to fully judge its correctness. We understand that the output of a region is considered a (sigmoidal) function of total input, where total input is the sum of stimulus-related and endogenous input. Why is it such that "the isolated effect of sensory input […] can then be expressed as the first-order derivative of the sigmoid"? It seems to us that this derivative would be the effect of any input.

Thank you for pointing this out. We agree that the description of the model was unclear. The total input S on the x-axis of Figure 5A can come in two forms: pre-stimulus activity (endogenous, Sn) and sensory stimulus (exogenous, Sx). The total output is expressed in the sigmoidal function O(Sn + Sx). However, because the amount of activity induced by the sensory stimulus itself (Sx) is assumed to be more or less constant from trial to trial over the physiological range of S, the isolated effect of the stimulus on the output – given a certain level of Sn – is expressed in the first-order derivative: O(Sn+Sx)–O(Sn))/Sx. As a result, the stimulus-evoked response as measured through gamma is proportional to the first-order derivative of the total output: O(Sn+Sx)−O(Sn)]/Sx. This is referred to as the gain, and is a function of pre-stimulus synaptic activity Sn (alpha-power). We have completely rewritten the section explaining this reasoning in the subsection “Pre-stimulus alpha power mediates cortical gamma responses”.

This mistake is a symptom of the authors sometimes conflating gain (which we would equate with the slope of the output/input curve) and actual input or output. Relatedly, the authors write "stimulus-related output gain" where they actually mean "output"; i.e. it is (if we understand correctly) precisely the output that is not stimulus-related which is relevant here, namely the endogenous fluctuations. There is confusion between gain, input, and output in some other places in this description as well; also how these terms map onto experimental measures is a bit ambiguous. (Gain = liberal vs. conservative; input = alpha; output = gamma? This is what we understand, but at times it appears as though alpha is equated to gain instead of input.)

Thanks for pointing this out – we apologize for the fact that the term definitions in the manuscript were not always consistent and at times confusing. We have now revised the entire manuscript, including the model description, with clear definitions in mind. Specifically, we now only refer to gain as the steepness of the sigmoid input-output curve (which is reflected directly in its derivative, describing the alpha-mediated effect of input (the sensory stimulus) on the output (gamma). Input to visual cortex is defined as the sum of endogenous activity (measured in alpha) and stimulus-related input (not measured, but assumed to be constant across trials). Finally, the alpha-mediated output of visual cortex is thought to be reflected in gamma activity.

7) The 3-way ANOVA reported in the last paragraph of the subsection “Alpha suppression enhances the gain of cortical gamma responses” is, we believe, not the correct way to analyze these data. The dependent variable here is gamma power, with independent variables condition (liberal/conservative) and alpha power bin (10 levels). Thus, a 2-way RM-ANOVA would be appropriate. If the authors believe the 3-way approach is indeed correct, then they should explain why this is so.

We fully agree with this comment and performed the suggested 2-way repeated measures ANOVA. Moreover, while revising the input-output model we realized that the U-shaped relationship between alpha and gamma predicted by the model is more appropriately described by a Gaussian-shape (this is explicit in the original R&D model, and in our depiction in Figure 5B) rather than a quadratic function (as used in the original submission). Therefore, we took the standard Gaussian with unit standard deviation and used this shape as the contrast of interest in the ANOVA instead of the standard quadratic contrast. Indeed, this uncovers an alpha bin-by-condition interaction effect at significance, suggesting that the input-output curves for liberal and conservative conditions show differential fits to a Gaussian shape. We describe the results of this analysis in the fourth paragraph of the subsection “Pre-stimulus alpha power mediates cortical gamma responses”.

8) The approach taken for the "within-subject group regression" is unclear to us (also not explained in Materials and methods). The primary evidence that links gamma activity to DDM drift bias is, it appears, based on regressing drift bias onto gamma power across different alpha bins, where both variables are averaged within bin, across participants (Figure 6). The correct approach here would be to perform this regression per participant, and then test whether the regression coefficients are different from zero at the population level. (Or better yet, show the individual regression lines.)

Indeed, we previously performed the regression analysis of drift bias on gamma after averaging the data within bins across participants. Although averaging across participants before regression suppresses noise (i.e. interindividual variability) by focusing on the within-subject group effect (see e.g. Linkenkaer-Hansen et al., 2004, and Kloosterman et al., 2015, for applications of this analysis), it required the extra step of normalizing the data of each of the individual participants (z-scoring) to prevent the possibility that the presumed within-subject correlation was actually driven by interindividual differences (i.e. by subjects with weak or strong responses at the group level). On the other hand, fitting regression lines per participant across the ten bins and testing the regression coefficients against zero is suboptimal because just a single outlier in a given participant’s ten bins can greatly affect the slope of the regression line for that participant, which reduces sensitivity of a subsequent statistical group-level test on the individual slopes. To avoid these issues, we instead performed a so-called repeated measures correlation with our data, a term coined by Bakdash and Marusich, 2017, for an approach originally introduced by Bland and Altman, 1995. This mixed-effects approach entails a correlation across all repeated observations from all participants, while controlling for differences in individuals’ average responses, and correcting the degrees of freedom of the correlation for the number of subjects. Because such a mixed-effects repeated measures correlation takes the non-independence of the ten observations per participant into account, it tends to yield greater statistical power than data that are averaged in order to meet the assumption of data point independence for simple regression/correlation (see Bakdash and Marusich, 2017, for more details). Finally, this approach allows us to directly enter the drift bias and alpha-binned gamma values into the correlation analysis without additional normalization steps, thereby aiding transparency of our analyses. Indeed, this more sensitive approach reveals highly significant positive correlations between gamma and drift bias in both conditions, which we report in our new Figure 6 and the accompanying text in the subsection “Visual cortical gamma activity predicts strength of evidence accumulation bias”.

9) The description of report proportions (Figure 2B) is not clearly defined. Shouldn't these sum to 1 within a condition?

Thank you for pointing this out. The reason they do not sum to one within a condition is that the hit rates and false alarm rates are computed using different subsets of trials: target present and target absent trials, respectively. Thus, the hit rate in Figure 2B is computed as the N hits/N target *present* trials, whereas the false alarm rate is computed as N false alarms/N target *absent* trials. Hit and miss rates do indeed sum to 1 since they are each other’s complement, and the same applies to false alarm and correction rejection rates, but hit rate and false alarm rate (by definition) do not sum to 1. We now point out the complements of the hit and false alarm rates in the legend of Figure 2B to make this explicit.

Additionally, it would be good to have some feeling of absolute number of responses, including those responses counting as a miss/correct rejection.

Thank you: we now report median trial counts across participants for all four signal-detection-theoretic trial categories in both conditions in the second paragraph of the subsection “Manipulation of decision bias affects sensory evidence accumulation”.

[Editors' note: further revisions were requested prior to acceptance, as described below.]

[…] The following weaknesses were noted and should be addressed directly. Please note that there is no guarantee of acceptance.1) The fact that the occipital alpha power is not clearly the decisive element here, but rather seems to be the consequence of a different signal of currently unknown (and still clearly under-investigated) origin that actually mediates the "drift bias". The idea that these dynamics are predominantly taking place within the visual system – as suggested in title and Abstract – is not sufficiently justified by evidence. This mostly amounts to a matter of emphasis – you do acknowledge that this "remains an open question" and that alpha modulation is not the whole story. Having said that, it is clear that the text (title Abstract and elsewhere) can be interpreted as trying to make such a point. So, I would suggest that a further rewrite on this point is in order, perhaps adding some of the details in the response-to-reviewers document to the manuscript (Discussion/supplement) itself.

Thank you for this suggestion. To once more investigate the existence of a top-down signal, we decided to further inspect the literature to determine what could be the source of the alpha effect. This prompted us to investigate the potential role of theta oscillations that mediate cognitive control mechanisms as a signature of top-down processes reflecting our experimental task manipulations. We were pleasantly surprised to find that when we took a wider pre-stimulus time window than we had done before (-1 to 0 seconds, necessitated by the wish to find a signature for theta) and performed a cluster-based permutation test over electrodes and frequencies (1-35 Hz), we not only recovered the alpha signal that we initially obtained by looking at the occipital electrode pooling only, but we also observed a clear modulation of pre-stimulus theta power (2-6 Hz) in midfrontal electrodes, with stronger theta in the liberal than in the conservative condition. We now highlight both findings in Figure 5. Further, we have made substantial changes throughout the manuscript to reflect the fact that the effect of our experimental manipulation is not only visual in nature.

One of the reviewers noted that you stated in your response that you updated the description of the Rajagovindan and Ding model and now account for the fact that an additional signal is required to explain these results – but the detailed explanation of this is lacking in the central description of this model in the subsection “Pre-stimulus alpha power mediates cortical gamma responses” (the second paragraph is not clear to understand and hides the fact that another external factor is needed here).

Thank you. Although we point to the necessity of an additional mechanism in other parts of the manuscript, we now also make the need for an additional mechanism clear in the description of the model in the Results subsection “Pre-stimulus alpha power mediates cortical gamma responses”.

2) Given the debate about whether you have identified the critical neural mechanism/correlate of the behavioral effect, it is even more important, for the potential novelty and significance of the results, to demonstrate that your results are specific to drift bias as opposed to starting point. This distinction is theoretically important, but the main conclusion in support of drift bias account is based on model fit metrics (BIC) alone. To be more confident that this difference is meaningful it is important to provide evidence that the drift bias model empirically captures the RT distributions better than the starting point model. In theory the better BIC fit indicates the drift bias does capture RT distributions better, but sometimes models can fit better using BIC (or other such metrics) for nuisance reasons unrelated to the core theoretical distinction. You do show individual subject RT distribution fits in Figure 2—figure supplement 3 for the drift bias model which look quite reasonable, but you do not show the corresponding fits to the starting point model, and the main claim of the paper rests on the ability to distinguish these with your task design/models.

Indeed, our main conclusion that the drift bias model best explains our behavioral data were based on the BIC results, and supported by the observed link between stimulus-induced gamma and drift bias. Bayesian model comparisons do consistently identify the drift bias model as superior across participants (for 15/16 participants), but the actual BIC differences between the two models are relatively small. We now show the subject RT distributions in Figure 2—figure supplement 3 for the starting point model as well, but to the naked eye both models seem to provide similarly reasonable fits to the RT distributions. We thus turned to the EEG data to test this main conclusion more rigorously. Specifically, we investigated the time courses of established EEG signatures of decision formation at the levels of sensory encoding and motor responses. Following previous studies, we hypothesized that a starting point bias would be reflected in a difference in baseline activity between the conditions before onset of the decision process, as has been shown previously at the motor output level during perceptual expectation (de Lange et al., 2013) and speeded decision making (Afacan-Seref et al., 2018). Conversely, we predicted that a drift bias occurring during the process of sensory evidence accumulation would be reflected in a steeper slope of post-stimulus and pre-response ramping activity, as well as a higher peak amplitude following stimulus onset. We now provide these results in a new Figure 4. A number of observations we make in this figure are in line with the drift bias model. First, we compared the two conditions in the pre-trial baseline period by looking at raw power. Specifically, we inspect two types of signals: i) stimulus-related activity (gamma modulation and the SSVEP), and ii) motor-related EEG signatures in the left motor cortex (left-hemispheric beta (LHB) power and the event-related potential (ERP)) in left motor cortex around the time of the behavioral response (as also suggested by the reviewer below). For none of these signals we found a statistically meaningful difference between conditions in the pre-trial baseline activity, suggesting a similar starting point of evidence accumulation in both conditions. Next, we looked at post-stimulus activity. After trial onset, in contrast, both sensory signals as well as the motor-related signal evolved differently in the liberal compared to the conservative condition, as expressed in higher peak level and steeper slope in the liberal condition. Together, these findings provide converging evidence that participants responded to the bias manipulations by adjusting the rate of evidence accumulation toward signal presence, but not its starting point. We have added these new findings to a new Figure 4 in the manuscript and report them in the subsection “EEG power modulation time courses consistent with the drift bias model”. Please also see our descriptions of the specific effects in response to the reviewer’s points below.

3) Relatedly, you do provide evidence that stimulus activity in gamma is amplified in the liberal condition, which is consistent with a drift bias, and also that this correlates with the extent of drift bias across subjects. What would be nice here is if you showed that this correlation was specific to the drift bias model and that it was statistically more evident than the corresponding correlations with starting point bias (i.e., you could test if the stimulus gamma activity is similarly correlated with estimated starting point bias in the alternative model). This would provide another more specific test of the evidence for the link between EEG and behavior even without identifying the top-down mechanism.

Thank you for this suggestion. To test whether drift bias was more strongly linked to gamma than starting point, we regressed both bias parameters estimated per alpha bin (within the two respective models) on gamma. Crucially, we found that in both conditions starting point bias did not uniquely predict gamma when controlling for drift bias (liberal: F(1,124) = 5.8, p = 0.017 for drift bias, F(1,124) = 0.3, p = 0.61 for starting point; conservative: F(1,124) = 8.7, p = 0.004 for drift bias, F(1,124) = 0.4, p = 0.53 for starting point. This finding again suggests that the drift bias model outperforms the starting point model when correlated to gamma power. We report this finding in the first paragraph of the subsection “Visual cortical gamma activity predicts strength of evidence accumulation bias”.

To sum up, we now present three independent and converging data points indicating that participants implement decision bias by adjusting the process of evidence accumulation, but not its starting point:

1) BIC values are lower for the drift bias model than for the starting point drift diffusion models.

2) Established sensory and motor-related EEG signals show no significant differences in pre-stimulus baseline activity in the liberal compared to the conservative condition, but do show stronger post-stimulus modulation.

3) The correlation between alpha-binned gamma and drift bias is robust to controlling for alpha-binned starting point.

We feel that these converging pieces of evidence together make a compelling case for our conclusion that decision bias is linked to adjustments in sensory evidence accumulation.